# Screening and Immune Efficacy Evaluation of Antigens with Protection Against Feline Calicivirus

**DOI:** 10.3390/vaccines12111205

**Published:** 2024-10-24

**Authors:** Yupeng Yang, Ruibin Qi, Mengru Chen, Kexin Feng, Zhe Liu, Hongtao Kang, Qian Jiang, Liandong Qu, Jiasen Liu

**Affiliations:** 1State Key Laboratory for Animal Disease Control and Prevention, Harbin Veterinary Research Institute, Chinese Academy of Agricultural Sciences, Harbin 150069, China; 2College of Veterinary Medicine, Northeast Agricultural University, Harbin 150030, China

**Keywords:** feline calicivirus, genotypes, recombinant protein, sneutralization titers, effectively protected

## Abstract

Background: Feline calicivirus (FCV), a pathogen that causes upper respiratory tract diseases in felids, primarily leads to oral ulcers and various respiratory symptoms, which can be fatal in severe cases. Currently, FCV prevention and control rely primarily on vaccination; however, the existing vaccine types in China are mainly inactivated vaccines, leading to a single prevention and control method with suboptimal outcomes. Methods and Results: This study commences with a genetic evolution analysis of Chinese FCV isolates, confirming the presence of two major genotypes, GI and GII with GI emerging as the dominant form. We subsequently selected the broadly neutralizing vaccine candidate strain DL39 as the template for the truncation and expression of multiple recombinant proteins. Through serological assays, we successfully confirmed the optimal protective antigen region, which is designated CE_39_ (CDE). Further investigation revealed the location of the optimal protective antigen region within the CE region for both the GI and GII genotype strains. Capitalizing on this discovery, a bivalent recombinant protein, designated CE_39_-CE_FB_, was generated. Cat antisera generated against CE_39_ and CE_39_-CE_FB_ proteins were used in cross-neutralization against various strains of different genotypes, yielding high neutralization titers ranging from 1:45 to 1:15 and from 1:48 to 1:29, respectively, which surpassed those induced by antisera from cats vaccinated with Mi-aosanduo (commercial vaccine, strain 255). Ultimately, in vivo challenge experiments were per-formed after immunizing cats with the CE_39_ and CE_39_-CE_FB_ proteins, utilizing Miaosanduo as a control for comparison. The results demonstrated that immunization with both proteins effectively made cats less susceptible to FCV GI, GII, and VSD strains infection, resulting in superior immune efficacy compared with that in the Miaosanduo group. Conclusion: These results indicate that this study successfully identified the antigen CE_39_, which has broad-spectrum antigenicity, through in vivo and in vitro experiments. These findings pre-liminarily demonstrate that the optimal protective antigen region of FCV strains is the CE region, laying a theoretical foundation for the development of novel broad-spectrum vaccines against FCV disease.

## 1. Introduction

Feline calicivirus (FCV) is one of the most common pathogens causing upper respiratory tract diseases in felines, primarily resulting in oral ulceration and various respiratory symptoms, and in severe cases, death [1,2]. It belongs to the *Vesivirus* genus of the *Caliciviridae* family and is a non-enveloped, single-stranded positive-sense RNA virus with a genome of approximately 7.7 kb [3,4]. The genome contains three open reading frames (*ORFs*), specifically *ORF1*, which is located at positions 20 to 5308 nt; *ORF2*, which spans from 5314 to 7317 nt; and *ORF3*, which is located between 7317 and 7634 nt [5,6]. The *ORF2* gene encodes the major capsid protein VP1, which is divided into six regions, A to F, and serves as the primary antigenic protein of FCV [7]. Analysis of different domains reveals that variations in the C, D, and E regions impact the antigenicity of FCVs [8]. The E region is highly prone to amino acid mutations, and the amino acid residues at positions 475–479 within this region can react with positive sera from cats infected with other FCV strains, indicating a high degree of conservation in this fragment [9,10]. Furthermore, studies have indicated that while the C and E regions of the FCV capsid protein are variable, most B-cell epitopes are concentrated in these regions, which has been further confirmed through pepscan epitope mapping [11].

Because the FCV RNA polymerase lacks a sequence correction function, the virus strain exhibits a remarkably high mutation rate and diverse antigenic profiles, leading to the emergence of different genotypes and gene subtypes despite their classification under the same serotype [12,13,14]. Currently, diseases caused by FCV infection in feline species have been reported in numerous countries worldwide, demonstrating a widespread distribution [15,16]. In China, multiple regions have reported incidents of FCV infecting feline species in recent years [17,18,19]. In addition, the virulent systemic disease FCV (VSD-FCV) is capable of infecting feline animals and causing systemic diseases that ultimately lead to their death; this disease is reported to occur in some countries [20,21,22,23,24,25]. Notably, FCV has been demonstrated to infect a wide range of feline species, including tigers, cheetahs, and lions, and reports indicate that dogs can serve as hosts for FCV [26,27,28,29,30].

In order to control the occurrence of FCV disease, the primary measure is vaccination [31]. FCV vaccines are categorized into attenuated live vaccines and inactivated vaccines [32,33]. In China, inactivated vaccines are utilized primarily for prevention and control [19], but there are no commercialized novel vaccines available at present. It is reported that due to the large antigenic difference among FCV strains, the neutralizing titers of existing inactivated vaccines against heterologous FCV strains are relatively low [34,35]. Therefore, the development of novel vaccines with broad-spectrum characteristics for the prevention and control of FCV diseases is quite important, and the most crucial aspect of novel vaccine development lies in the screening of antigenic epitopes. In this study, we employed serological testing to identify a broadly protective antigenic region, designated the DL39-CDE region (CE_39_), and verified that the optimal protective antigen for FCV strains of different genotypes is the CE (CDE) region. The potential of the protein prepared from this antigenic region to be used as a subunit vaccine was further investigated, and the results demonstrated a good protective effect. Overall, this research provides technical support for the further development of novel vaccines against FCV.

## 2. Methods

### 2.1. Cells and Viruses

F81 cells were stored in our laboratory and cultured in Dulbecco’s modified Eagle’s medium (DMEM; Gibco, Fort Worth, TX, USA) supplemented with 10% fetal bovine serum (FBS; Gibco, USA), 100 U/mL penicillin, and 100 µg/mL streptomycin (Biosharp, Hefei, China) at 37 °C under 5% CO_2_. The DL31 (GenBank accession number MW804427), DL38 (MW804429), DL39 (MW804430), HRB48 (MW804434), TIG-1 (KU373057), and FB-NJ-13 (KM111557) strains of FCV were stored in our laboratory. Cat anti-DL39 and FB-NJ-13 sera were preserved in our laboratory. The pET-32a expression plasmid was also maintained in our laboratory.

### 2.2. Phylogenetic Analyses of the FCV Whole-Genome

Publicly available whole-genome nucleotide sequences of FCVs (including 61 domestic FCV strains circulating in China as well as F9 and FCV 2280 strains) were retrieved from GenBank. The downloaded whole-genome sequences were subjected to nucleotide sequence alignment analysis via MEGA-X software (10.1.8) with adjustments made accordingly as previously described [36]. A maximum likelihood (ML) phylogenetic tree was subsequently constructed using the complete nucleotide sequences with specific conditions indicated in the figure captions. Further modification of the phylogenetic tree was conducted via https://www.chiplot.online/ (accessed on 6 August 2024) [37].

### 2.3. Construction of Truncated Recombinant Plasmids

To screen for antigenic regions with broad-spectrum properties, segmented protein expression primers were first designed on the basis of the VP1 sequence of the FCV DL39 strain (GI), which could provide broad-spectrum protection according to our previous study (the structural diagram of the segmented design is illustrated in Figure 1). These primers targeted specific regions of VP1, including the B, CD, E, F, B–D (BD), and C–E (CE) regions (Appendix A). Additionally, to further clarify the optimal protective antigenic regions of the genotype GII strain (FB-NJ-13 strain), similar procedures were performed (Appendix A). Primer Express 3.0 software was used to design the necessary amplification primers for fragment amplification, and the homology arms and restriction sites of the pET-32a vector were included in the primers (Appendix A). The primer design is detailed in Appendix A. Both the amplification products and the constructed plasmids were sequenced to ascertain their accuracy. The correctly amplified products were then ligated into the pET-32a vector for subsequent protein (FCV antigenic region with a TrxA tag and His-tag) expression. Even though all different regions of FCV VP1 were successfully expressed in BL21-competent cells, the structures of the different subunits after expression have not yet been checked.

### 2.4. Expression and Purification of Truncated Recombinant Proteins

The expression of the aforementioned recombinant plasmids was induced, and the expressed proteins were designated B_39_, CD_39_, E_39_, F_39_, BD_39_ (BCD), CE_39_ (CDE), B_FB_, CD_FB_, E_FB_, F_FB_, BD_FB_ (BCD), and CE_FB_ (CDE). Initially, bacterial cultures containing each recombinant plasmid were transferred to 10 mL of 2YT medium (containing 100 μg/mL ampicillin) at a ratio of 1:100. The IPTG concentration, expression temperature, and expression time were optimized separately for each plasmid, and protein expression was conducted under the optimal conditions. Following expression, the bacterial cultures were sonicated, and the recombinant proteins were purified via the HisSep Ni-NTA Agarose Resin His Kit. The concentration of the purified proteins (FCV antigenic region with a TrxA tag and His-tag) was then determined with BCA Protein Assay Kit of Beiotime Biotechnology. Finally, the expressed and purified recombinant proteins were analyzed and identified through SDS–PAGE and Western blot (Wb) assays. During the Wb validation process, His-tagged antibody and mouse anti-FCV 2280 serum were utilized as primary antisera for the purpose of confirming specificity.

### 2.5. Competitive ELISA of Recombinant Proteins

A competitive ELISA was performed using purified recombinant proteins and homologous virus-derived cat antisera (anti-DL39 and anti-FB-NJ-13 sera) to preliminarily screen for the optimal antigen. Initially, the FCV DL39 and FB-NJ-13 strains were diluted to 1 × 10^6^ TCID_50_/mL and added to the ELISA plate at 100 μL/well, which was followed by incubation overnight at 4 °C. After three washes with Tris Buffered Saline with Tween-20 (TBST, TBS containing 0.05% Tween-20), antigen–antibody reactants were prepared from the cat antisera and their corresponding recombinant proteins. The sera were diluted according to the optimal dilution ratio of 1:100 for the ELISA assay using FCV cat anti-positive serum (diluent is Phosphate Buffer Saline (PBS)), and a fresh ELISA plate was prepared. Each well was subsequently loaded with 0.1 nmol of each protein and 100 μL of the diluted serum, which was thoroughly mixed, and triplicate reactions were conducted for each sample. The resulting mixtures were incubated in a constant-temperature incubator at 37 °C for 1 h. Following the completion of the reaction, each well was washed by TBST, and the reaction mixture was added as the primary antibody to the reaction plate containing the coated antigen and further incubated in a constant-temperature incubator at 37 °C for an additional hour. Concurrently, a mixture of purified TrxA protein and cat antisera was added to the control wells. Upon completion of the reaction, a commercially available HRP-labeled goat anti-cat secondary antibody was added, and the mixture was incubated in a temperature-controlled incubator at 37 °C for 1 h. Next, 100 μL of TMB chromogenic solution was added, and color development was allowed to proceed in an incubator maintained at 37 °C for 15 min. The reaction was stopped with 2 M H_2_SO_4_, and the absorbance was measured at 450 nm.

### 2.6. Competitive Neutralization Assay with Recombinant Proteins

#### 2.6.1. Cat Antisera Raised Against DL39 and FB-NJ-13 Used in Neutralization Assays Against Prevalent FCV Strains

A neutralization test was performed utilizing the sera of the laboratory-preserved FCV DL39 and FB-NJ-13 strains to quantify their neutralization titers against the prevalent FCV strains. The objective of this evaluation was to verify that the sera of the two strains met the criteria necessary for a competitive neutralization assay.

#### 2.6.2. Competitive Neutralization Assay Between Recombinant Proteins and the Homologous Strains

Using a competitive neutralization assay, proteins with the optimal neutralizing activity were screened. The principle of the competitive neutralization test is based on antigen competition. Initially, the protein is co-incubated with positive serum. If this protein binds to the neutralizing antibodies in the serum, the subsequent binding of the serum to the virus will result in a reduction in the neutralization titer. The final results are expressed in terms of neutralization rate. A lower neutralization rate indicates a stronger ability of the protein to neutralize the serum, theoretically making it an optimal protective antigen. First, the optimal serum dilution was determined. Preliminary experiments revealed that a 1:8 dilution of the sera targeting the two virus strains resulted in the most favorable conditions for the competitive neutralization assay. A nanomolar gradient ranging from 10^0^ to 10^−6^ nmol was subsequently established for the recombinant protein with increases of 10-fold at each step. Each concentration was replicated five times. Equivalent volumes of the respective diluted homologous virus sera were then added and incubated together in a constant-temperature incubator set at 37 °C for 1 h. After incubation, the diluted homologous virus associated with each recombinant protein was added to the mixture, thoroughly mixed on a shaker, and then incubated again in a constant temperature incubator at 37 °C for 1 h. After 1 h, the residual liquid in the 96-well plate containing the cells was discarded. The mixture was added to five replicate wells, and the samples were labeled with the time and virus strain name. The cell status was observed every 12 h, and the number of damaged cells in each well was recorded. The results were calculated via the Reed–Muench method.

#### 2.6.3. Competitive Neutralization Between Recombinant Proteins and Different FCV Strains

Through the aforementioned competitive neutralization test, we statistically analyzed the final neutralization rates under different concentrations of each protein. The protein with the lowest neutralization rate was selected as the optimal protective protein for subsequent experiments. The respective recombinant proteins screened from the two serum samples were co-incubated and then subjected to competitive neutralization assays against different strains such as HRB48, TIG-1, DL38, and 2280, following the procedures described in Section 2.6.2.

### 2.7. Preparation of Bivalent Recombinant Proteins Based on Different Genotypes

The CE fragments of VP1 from the FCV DL39 and FB-NJ-13 strains were amplified via the primers in Appendix A, resulting in a band size of 393 bp (Appendix A). After successful amplification, fusion PCR was performed on the two fragments. The recombinant proteins were expressed after sequencing verification. The optimal expression conditions for the recombinant protein were determined, and protein expression was conducted under these conditions. The recombinant protein was subsequently purified and further detected via SDS–PAGE and Wb assays.

### 2.8. Challenge Test with FCVs of Different Genotypes

Utilizing the optimized protective antigens that were identified by screening, Montanide™ GEL was employed as an adjuvant in conducting immunization and challenge tests with the recombinant proteins. The selection of strains was based on factors such as the isolated host, virulence, and phylogenetic analysis, ensuring the inclusion of representative strains. The strains chosen for the experiment included strain HRB48 (GI), strain FB-NJ-13 (GII), and the standard VSD strain 2280 (GI, VSD). A schematic diagram is presented in Figure 2.

Thirty-nine 2-month-old Chinese domestic cats were prepared. Initially, the FCV antibodies of all the cats were detected via the ImmunoComb^®^ Feline VacciCheck Antibody Test Kit for Feline Calicivirus, Herpesvirus, and Panleukopenia Virus. After confirming that all cats were negative for FCV antibodies, they were randomly divided into 13 groups (Groups 1–3 were immunized with the CE_39_, Groups 4–6 were immunized groups with CE_39_-CE_FB_, Groups 7–9 were immunized with the Miaosanduo commercial vaccine (FCV-255), Groups 10–12 were regarded as positive groups, and Group 13 was the negative control group) with 3 cats in each group. The cats were placed in separate cages to prevent cross-infection in the later stages. The cats were immunized with the recombinant protein mixed with Montanide™ GEL adjuvant (Seppic, France)at a ratio of 1:1. Each cat received 50 μg of the recombinant protein through subcutaneous injection in the neck. The positive control group and negative control group received the same volume of the mixture DMEM with Montanide™ GEL (1:1) for immunization. After the initial immunization, a booster dose of the same amount was administered 21 days later (second immunization). On the 14th day after the second immunization, 200 μL of cat antisera was harvested for preliminary neutralization testing, which is a primary method for assessing vaccine efficacy. Furthermore, on the same day, with the exception of the negative control group, the corresponding groups underwent viral challenge via nasal instillation at a dose of 1 × 10^8^ TCID_50_ (Groups 1, 4, 7 and 10 were challenged with HRB48; Groups 2, 5, 8 and 11 were challenged with FB-NJ-13; Groups 3, 6, 9 and 12 were challenged with 2280), whereas the negative control group (Group 13) received an equivalent volume of DMEM. After viral challenge, the clinical manifestations of each group were rigorously monitored and scored at various time points. The clinical scoring evaluation criteria were devised through a synthesis of European standards and specific experimental conditions (Appendix A). Drawing upon the empirical observations from the animal trials, we categorized instances with a clinical score of 4 or below, devoid of oral ulceration, as negative outcomes. Body temperature, body weight, and viral excretion in oral and anal swabs were measured every two days. Simultaneously, whole blood samples (300 μL) were collected at the same frequency for viremia assessment. Utilizing the FCV-fluorescence quantitation methodology devised by our laboratory, we conducted virus load detection in viral excretion and viremia. Detailed daily clinical symptom records were maintained for the cats in each group with a minimum of two individuals present during each recording session. Samples were systematically collected every two days and promptly processed, ensuring appropriate storage conditions. Subsequent to the completion of the testing, all samples underwent uniform treatment prior to further analysis. Upon the conclusion of the experiment, all experimental cats were humanely euthanized. Pathological alterations in the lungs and visceral organs were subsequently examined through dissection, and pathological tissue sections were prepared for further observation.

### 2.9. Statistical Analysis

The statistical analysis was performed using GraphPad Prism 8.0 software. Comparisons were made using either a t-test or two-way ANOVA. Specifically, “ns” denotes no significant difference, and “*” denotes significant difference (“*”, *p* < 0.05; “**”, *p* < 0.01; and “***”, *p* < 0.001).

## 3. Results

### 3.1. Phylogenetic Analysis of the FCV Whole-Genome Sequence

The phylogenetic analysis of the full-genome nucleotide sequences of FCV (isolated from China) underscores the wide spread of FCV strains across multiple regions in China, which can be distinctly categorized into two genotypes: GI and GII. Among the currently circulating strains, a preponderance of 68.9% (42/61) belong to the GI genotype, including both the vaccine strain F9 (M86379) and the virulent strain FCV 2280. Conversely, 31.1% (19/61) of the strains are classified under the GII genotype with a noteworthy observation that all leopard-derived strains exclusively fall within this genotype (Figure 3). Furthermore, the analysis revealed that FCV strains isolated from the same geographical region may exhibit genetic diversity and belong to distinct evolutionary branches (Figure 3). These findings underscore the dominance of the GI genotype among prevalent strains in China yet emphasize the need for vigilance against the potential hazards posed by GII genotype strains.

### 3.2. Expression and Purification of Recombinant Proteins

The recombinant protein fragments were amplified via the primers listed in Appendix A, subsequently ligated into vectors, and expressed as proteins. To achieve the efficient expression of these recombinant proteins, we optimized the concentration of IPTG, expression temperature, and expression duration and combined these conditions to develop a method capable of efficiently expressing the recombinant proteins (Appendix A). Coomassie blue staining revealed that the purified recombinant proteins exhibited single bands of the expected sizes (Figure 4a). The specificity of the purified proteins was verified by Wb assays using anti-His and anti-FCV 2280 mouse antisera as primary antibodies. The results demonstrated specific bands corresponding to the target proteins, with no additional bands observed, confirming the high purity and specificity of the purified proteins (Figure 4b,c). Notably, the anti-His antibody successfully detected the Trxa protein, whereas the FCV mouse antiserum did not, further validating the specificity of the purified recombinant proteins. The concentrations of the purified recombinant proteins were determined (Appendix A).

### 3.3. Results of Competitive ELISA with Recombinant Proteins

The molar concentrations of the recombinant proteins were calculated on the basis of their measured concentrations, and nanomolar quantities were used to conduct competitive ELISA experiments. Following a 1 h co-incubation of the purified recombinant proteins with sera prepared from their respective homologous virus strains, significant reductions in ELISA OD_450_ values were observed. Notably, no significant difference was detected between the TrxA protein group and the NC group, indicating that the TrxA tag attached to the recombinant proteins did not affect the experimental results (Figure 5). In the validation experiments conducted on strains FCV DL39 (GI) and FB-NJ-13 (GII), the role of the CE protein was the most prominent, with OD_450_ values of 0.289 and 0.45, respectively. These values exhibited significant differences compared to the control group with the TrxA protein (*p* < 0.001). Additionally, the BD and F proteins also showed notable competitive effects compared with the TrxA protein (Figure 5a,b). These results preliminarily suggest that the CE, BD, and F proteins of both GI and GII strains have strong reactivity, which can serve as an important basis for screening the optimal protective antigenic epitopes.

### 3.4. Results of the Competitive Neutralization Assay with Recombinant Proteins

In the present investigation, we initially ascertained that the feline antiserum directed against both the FCV DL39 and FCV FB-NJ-13 strains possessed the ability to elicit neutralizing titers exceeding 1:16 against diverse strains, rendering FCV DL39 and FCV FB-NJ-13 suitable for subsequent experiments (Appendix A). Competitive neutralization assays were conducted using recombinant proteins and their corresponding homologous virus cat antisera with the neutralization rate of the cells serving as the outcome metric. A lower neutralization rate indicates better protein performance. The results revealed that the CE_39_ and E_39_ proteins exhibited significantly lower neutralization rates compared to other proteins, indicating that they are the main targets of neutralizing antibodies, and CE_39_ showed an even more pronounced effect (Figure 6a). In the context of the genotype GII strain FB-NJ-13, the CE_FB_ and E_FB_ proteins also displayed significantly reduced neutralization rates compared with alternative proteins, indicating that the CE_FB_ and E_FB_ proteins of the GII strain are the main targets of neutralizing antibodies (Figure 6b). To validate the broad-spectrum potential of the CE_39_, E_39_, CE_FB_, and E_FB_ proteins, additional competitive neutralization assays were conducted. Upon mixing these four proteins with sera derived from their respective homologous viruses, they effectively diminished the cross-neutralizing titers of these sera against other strains. As the protein concentration increased, the neutralizing titers progressively decreased to zero with CE proteins exhibiting a more rapid reduction (Figure 7). These findings suggest that among the tested proteins, the CE protein is the optimal neutralizing antigen.

### 3.5. Preparation of Divalent Recombinant Proteins

The primer pairs listed in Appendix A were used to amplify the recombinant protein fragments CE_39_ and CE_FB_, and then fusion PCR was performed. The resulting products were subsequently ligated into a vector for protein expression (Appendix A). To ensure optimal protein expression, the conditions were carefully optimized, identifying an IPTG concentration of 0.2 nmol/μL, an expression temperature of 16 °C, and an expression duration of 18 h. Under these conditions, the recombinant proteins were successfully expressed. Following expression, the proteins were purified, and their purity was analyzed via SDS–PAGE. The results demonstrated that the target proteins were pure and consistent with the expected size (Figure 8a). To further verify the specificity of the purified proteins, Wb assays were performed using both anti-His as the primary antibody and FCV 2280-specific mouse antiserum as the primary antibody. Both assays confirmed the purity and size consistency of the target proteins (Figure 8b,c). The concentration of the purified CE_39_-CE_FB_ protein was 526 ng/μL.

### 3.6. Results of Challenge Test with FCVs of Different Genotypes

#### 3.6.1. Results of the Immunogroup Cross-Neutralization Tests

Serum samples were collected for cross-neutralization tests, which revealed that all three immunization groups (CE_39_, CE_39_-CE_FB_, and Miaosanduo) generated neutralizing titers against genotype GI (HRB48, 2280, and TIG-1) and genotype GII (DL38 and FB-NJ-13) strains. Notably, the CE_39_ and CE_39_-CE_FB_ immunization groups exhibited higher neutralizing titers against prevalent strains in China, ranging from 1:45 to 1:15 and from 1:48 to 1:29, respectively. In contrast, the Miaosanduo immunization group displayed lower neutralizing titers compared to the other two groups against the prevalent Chinese strains DL38, HRB48, and FB-NJ-13 (Figure 9).

#### 3.6.2. Results of Animal Challenge Tests with the FCV Genotype GI Strain

Statistical analysis was conducted on the results with Group A representing the CE_39_ immunization group, Group B the CE_39_-CE_FB_ immunization group, Group C the Miaosanduo immunization group, Group D the HRB48 virus positive group, and Group E the negative control group. A challenge test was performed using the FCV genotype GI strain, HRB48. The results indicated that all cats of the positive group presented typical clinical symptoms of FCV infection, including weight loss, fever, depression, dyspnea, coughing, increased ocular secretions, and oral ulcers. In the CE_39_ immunization group, two cats displayed symptoms such as depression, fever, and weight loss, whereas one cat presented with ocular secretions and mild respiratory distress. In the CE_39_-CE_FB_ immunization group, one cat manifested depression and weight loss, and another cat had ocular secretions. The Miaosanduo group uniformly showed symptoms of depression, fever, and continuous weight loss with one cat additionally experiencing ocular secretions and mild respiratory distress (Figure 10b,c). However, none of the three immunization groups exhibited typical oral ulcers. Virus shedding and viremia in the three immunization groups were lower than those in the virus positive group. However, virus shedding persisted in all three immunization groups for up to 14 days after challenge, and substantial amounts of virus shedding were still observed in all experimental groups by the end of the observation period (Figure 10d–f). Clinical scoring indicated that the recombinant protein CE_39_ and CE_39_-CE_FB_ immunization groups had a score of 2.67, whereas the virus positive group scored significantly higher at 12.33 (*p* < 0.001), indicating a superior immune effect of the recombinant proteins. Additionally, the Miaosanduo immunization group scored higher than the recombinant protein immunization groups, suggesting that the recombinant proteins provided better immune protection (Figure 10a). These findings demonstrate that the recombinant proteins CE_39_ and CE_39_-CE_FB_ confer an effective protection rate against the genotype GI strain HRB48 than Miaosanduo.

#### 3.6.3. Results of Animal Challenge Tests with the FCV Genotype GII Strain

A challenge test was conducted using the FCV genotype GII strain FB-NJ-13. The results revealed typical clinical manifestations of FCV in the positive group, including fever and weight loss, depression, dyspnea, severe coughing, and oral ulcers, and one cat developed leg ulcers and conjunctivitis. Among those in the CE_39_ protein-immunized group, three cats presented depression, two cats developed fever up to 39 °C, and one cat had mild dyspnea. In the CE_39_-CE_FB_ protein-immunized group, one cat presented with depression, and another presented with minimal ocular discharge, whereas no significant changes in body temperature or weight were noted. The Miaosanduo-immunized group universally displayed depression, fever, and continuous weight loss. Additionally, one cat in this group exhibited mild dyspnea and coughing (Figure 11b,c). However, none of the three immunization groups manifested typical oral ulcers. The excretion and viremia in all three immune groups were lower than those in the positive group with a reduced excretion load and viremia after viral infection. However, excretion persisted in the immune groups for up to 14 days after exposure (Figure 11d–f). Notably, the CE_39_-CE_FB_ immune group demonstrated superior performance in the challenge test with the FB-NJ-13 strain. Clinical scoring revealed significantly lower scores of 3 and 1.33 for the recombinant protein CE_39_ and CE_39_-CE_FB_ immune groups, respectively, than the score of 14 in the positive group (*p* < 0.001). Both scores were also lower than those of Miaosanduo, indicating stronger immune effects of the two recombinant proteins (Figure 11a). The results demonstrated that both recombinant proteins, CE_39_ and CE_39_-CE_FB_, provided effective protection against the genotype GII strain FB-NJ-13 than Miaosanduo.

#### 3.6.4. Results of Animal Challenge Tests with the FCV VSD Strain

The challenge test with the VSD strain (GI, 2280) revealed the emergence of typical clinical symptoms of FCV in the positive group, encompassing elevated body temperature, reduced body weight, lethargy, dyspnea, severe coughing, and oral ulcers. One cat died from the disease on the 7th day, whereas the remaining cats survived until the end of the experiment, albeit with clinical symptoms persisting until the 12th day. In the recombinant protein CE_39_ immunization group, two cats exhibited lethargy, and one showed mild respiratory distress. In the CE_39_-CE_FB_ immunization group, one cat displayed lethargy, and another presented with minimal ocular discharge. In the Miaosanduo immunization group, two cats had elevated body temperatures and two experienced respiratory distress (Figure 12b,c). All cats in this group were lethargic, but none of the three immunization groups manifested typical oral ulcers. Both viral shedding and viremia were lower in the three immunization groups than in the positive group with reduced viral loads and viremia levels (Figure 12d–f). The clinical symptom scores for the CE_39_ and CE_39_-CE_FB_ immunization groups were 1.67 and 2, respectively, which were significantly different (*p* < 0.001) from the challenge group score of 14.33. The Miaosanduo group had a score of 4.67, which was greater than that of the CE_39_ and CE_39_-CE_FB_ groups, preliminarily indicating that the two recombinant proteins provided stronger immune protection than Miaosanduo was able to provide (Figure 12a). The challenge-immunization test results demonstrated that both recombinant proteins, CE_39_ and CE_39_-CE_FB_, confer a more effective protection rate against the VSD strain than Miaosanduo.

#### 3.6.5. Results of Histopathological Analysis

After euthanasia, the lungs of cats from each group were harvested for the inspection of pathological changes and subsequent histopathological examination. The lungs and pathological tissues of the control group showed no abnormalities. Similarly, no typical lung lesions were observed in the CE_39_, CE_39_-CE_FB_, or Miaosanduo immunization groups, and the histopathological examination results revealed no significant pathological changes. In contrast, varying degrees of hemorrhage or congestion were observed in the lungs of the HRB48-, FB-NJ-13-, and 2280-positive groups. Histopathological examination further revealed a mild thickening of the alveolar walls in the HRB48-positive group, mild to moderate thickening of the alveolar walls and moderate thickening with partial alveolar lumen occlusion in the FB-NJ-13-positive group, and focal moderate thickening of the alveolar walls with partial alveolar lumen occlusion and alveolar macrophage proliferation and infiltration in the 2280-positive group (Figure 13). These results indicate that the recombinant proteins CE_39_ and CE_39_-CE_FB_ can effectively prevent lung pathological damage induced by different genotypes of FCV strains.

## 4. Discussion

Single-stranded positive-sense RNA viruses display an exceptionally high mutation rate, predisposing them to antigenic variation, posing significant challenges in the prevention and control of associated diseases. Presently, FCV has evolved an array of sophisticated strategies to circumvent the host’s immune defenses [38]. The presence of an error-prone viral polymerase in FCV facilitates the relentless accumulation of mutations within its genome, which in turn bolsters the virus’s adaptability within its environmental niche, ultimately culminating in immune evasion [7,38].

Currently, the prevention and control of FCV predominantly rely on inactivated and attenuated vaccines. The multitude of vaccines available on the market, including the FCV F9, FCV 21, and FCV-255 strains [34,39,40,41], belong to the realm of traditional vaccines, yet their efficacy remains a subject of debate. Reports in both domestic Chinese and international literature have indicated that existing vaccines may demonstrate suboptimal cross-neutralization capabilities against the currently prevalent viral strains [19,42]. Previous investigative reports focusing on upper respiratory disease cases revealed that in China, cats fully vaccinated with feline trivalent vaccines are still at a certain degree of risk of developing FCV-associated stomatitis. Despite the availability of newly commercialized inactivated vaccines developed in China to address local epidemiological situations, there is no doubt that the continuous evolution of FCVs poses new challenges for vaccine design [25,43]. Thus, research on novel vaccines for the prevention of this disease is vital. This research primarily encompasses the optimization of antigen design, the enhancement of protective efficacy, and the improvement of vaccine safety [44,45]. Among these, optimizing antigen design, which involves analyzing the genetic diversity of key viral antigen genes, serves as a pivotal foundation for the development of novel vaccines. The antigenic epitopes of FCVs are predominantly concentrated in VP1 [46,47]; the E region is the primary antigenic epitope domain containing multiple antigenic sites [9,10,11]. Research has revealed that the hypervariable region at the 5′ end of the E region and the C region are concentrated areas of neutralization sites, whereas the B, D, and F regions harbor non-neutralizing antibody sites [48]. Analysis of different domains has shown that the variability within the C, D, and E regions can influence the antigenicity of the virus [8].

At present, FCV vaccines capable of eliciting broad humoral immune responses have been developed [4,49], and the generation of extensive humoral immunity is also a crucial aspect in the development of novel vaccines. Previous findings have conclusively demonstrated that the FCV DL39 strain possesses robust broad-spectrum protective efficacy against the currently prevalent genotype GI and GII strains [14]. As a result, the present study capitalized on the previously screened DL39 strain to delve deeper into identifying the pivotal regions of protective antigens, laying a solid theoretical groundwork for the creation of novel vaccines.

Compared to indirect ELISA, competitive ELISA has certain limitations, including reduced sensitivity and increased operational complexity. Despite these inherent constraints, competitive ELISA demonstrates remarkable reproducibility and is an ideal methods for the identification and screening of viral antigen epitopes, having already been implemented in practical applications [50,51]. Consequently, this study utilized the competitive ELISA technique to measure antibody levels induced by the FCV structural protein VP1 in the host. It is important to note, however, that ELISA can only quantify the total antibody levels produced and cannot ascertain whether these antibodies exhibit neutralizing activity. For instance, this study employed competitive ELISA to illustrate that the BD, CE, and F regions of the FCV structural protein VP1 in both the DL39 and FB-NJ-13 strains exhibit significant immunogenicity, leading to the induction of high antibody titers. Nevertheless, it remains uncertain whether these antibodies possess neutralizing activity. Neutralization assays can be utilized to detect the neutralizing antibody levels produced by the body, and competitive neutralization assays can also determine the specific antigenic regions or peptide segments that generate neutralizing antibodies [52]. Using competitive ELISA and neutralization assays, we successfully identified the FCV DL39 CE_39_ region as the optimal antigenic location. Remarkably, the efficacy demonstrated by this CE_39_ region surpassed that of the previously documented individual E region [9,10,11], underscoring its potential significance in vaccine development. Furthermore, to extend our investigation, we conducted a screening for the optimal protective antigen region within the representative genotype GII strain FB-NJ-13. The outcomes revealed that the optimal protective antigen region was consistently located within the CDE domain and was shared across the genotype GI and GII strains. The prokaryotic expression of the protein derived from this region demonstrated potent competitive neutralizing capabilities, providing a solid foundation for the subsequent development of protective multivalent subunit vaccines and other innovative vaccine formulations.

An intriguing observation emerged during the screening process for identifying the optimal neutralizing antigen: CE_FB_ demonstrated a seemingly better competitive neutralization effect against the HRB48 (GI) strain (Figure 7a,c). Could this observation suggest that cross-protection in FCV may not be intrinsically tied to its genotype? Our results from cross-neutralization tests conducted that strains within the same genotype tend to indicate higher neutralization titers (Appendix A). However, such a conclusion cannot be definitively drawn due to the necessity of validating the findings with a more extensive range of strains. This underscores the limitation of competitive neutralization tests using a single neutralizing antigen, as they cannot definitively represent cross-protection among strains due to numerous constraining factors, including cross-neutralization titers. These considerations prompt us to contemplate that a multivalent vaccine incorporating various genotypes may represent an advantageous strategy. Therefore, to further bolster the protective efficacy, we synthesized a bivalent recombinant protein, designated CE_39_-CE_FB_, by concatenating protective antigens derived from distinct genotypes. Subsequently, we conducted immunoprotective assays to evaluate its performance. The outcomes revealed that both CE_39_ and CE_39_-CE_FB_ exhibited substantial protective effects against FCV, which aligns the findings of a recent study [53]. While we cannot definitively assert that our combination constitutes the optimal protective antigen combination, as assessing the optimality of a combination typically necessitates comparative validation against multiple other combinations, based on the current experimental results, we have successfully identified a protective antigen combination pattern that exhibits favorable performance across different genotypes.

In the course of the study, it was observed that the cross-neutralizing titers elicited by the recombinant proteins were lower than those induced by the inactivated vaccines. Homologous and heterologous FCV antibody responses were intimately linked to protection against FCV-induced clinical diseases [37]. It has been documented that the incorporation of immunostimulatory molecules enhances the immune response elicited by vaccines. Consequently, to address the aforementioned issues, the inclusion in the two recombinant proteins of reported immunostimulatory molecules that augment humoral and cellular immunity, such as IL-2, IL-4, and IFN-γ, could potentially fortify the immune response triggered by the vaccine upon administration [54,55,56]. Moreover, integrating bioinformatics and structural biology to guide the refinement of FCV protective antigen design is of crucial significance. This strategy involves the selection of antigen fragments capable of provoking potent and long-lasting immune responses, serving as the cornerstone of subunit vaccine development. Additionally, the pursuit of research endeavors focused on multi-epitope and polyvalent subunit vaccines, aimed at augmenting their broad-spectrum protective capabilities, is of great value.

Although the mortality rate of the classic FCV is not high, its threat to kittens cannot be overlooked [22,25]. FCV can replicate effectively within infected animals, resulting in various clinical symptoms such as oral ulcers [15]. Even after the recovery of clinical symptoms, it can still induce infected animals to shed the virus for up to two weeks or transform into a latent infection [57]. This study disclosed that the co-expression of DL39 and FB-NJ-13 CE regions could strikingly mitigate the clinical symptoms of FCV infection in cats, although typical oral ulcer symptoms were not manifested. Nonetheless, symptoms such as elevated body temperature and weight loss were still observable, which is accompanied by viral excretion. The suboptimal immune effectiveness could potentially be ascribed to the fact that the amino acid peptide segments expressed by the prokaryotic system might possess disparate advanced structures in comparison with the natural virus particle, leading to the loss of partial or complete conformational antigenic epitopes, or the swift release of antigenic short peptides, which failed to induce higher antibody levels prior to degradation [58]. To augment the protective potency of the subunit vaccine co-expressing the CE regions, our subsequent investigations will be centered on optimizing the amino acid sequence of the CE regions to better exhibit the antigenic epitopes and select more safeguarded and efficient immune adjuvants and cytokines to induce more efficacious humoral and cellular immune responses.

Considering that the majority of calicivirus members lack efficient in vitro proliferation and culture systems, this has posed substantial impediments to the development of inactivated or attenuated vaccines [59]. Moreover, owing to the high mutational characteristics of RNA viruses, there exist numerous subtypes of norovirus, and poor cross-immunogenicity exists among different subtypes, which has precluded the commercialization of norovirus vaccine products to this day [60,61]. The norovirus vaccines currently undergoing clinical evaluation are all multi-subunit virus-like particle vaccines based on insect cells [61]. Nevertheless, the excessive number of variant subtypes and the continuous evolution of the virus have rendered the selection of effective broad-spectrum antigenic epitopes as another effective strategy for preventing calicivirus infections. This study on the screening of the optimal epitope of FCV can further provide some theoretical references for other related research on calicivirus.

In this study, the potential feasibility of using subunit vaccines for FCV prevention and control was successfully demonstrated, and a broad-spectrum protective antigen region was identified that holds significant implications for the future design of novel vaccines, such as mRNA vaccines. In summary, this research has broadened the prospects for the prevention, control, and treatment of FCV diseases and has provided substantial safeguards for enhancing the welfare of feline animals.

## 5. Conclusions

In this study, an evolutionary tree was constructed utilizing the whole-genomic sequences of FCVs isolated in China, confirming that the prevalent viral strains in China are predominantly categorized into genotypes GI and GII. Subsequently, the neutralizing antigen FCV DL39 CE_39_ protein was successfully isolated and screened. Both in vivo and in vitro experimental assessments confirmed that this recombinant protein exhibited effective protective efficacy against strains belonging to both the GI and GII genotypes. Additionally, a recombinant bivalent protein, designated as CE_39_-CE_FB_, which encompasses antigens from both genotypes, was further developed. It similarly offers robust protection against strains of both the GI and GII genotypes.

## Figures and Tables

**Figure 1 vaccines-12-01205-f001:**
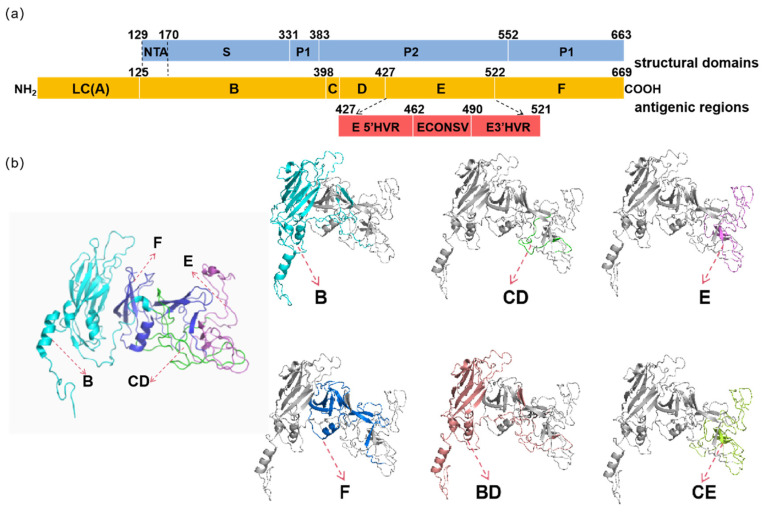
Schematic diagrams of protein structures. (**a**) Schematic diagram of VP1 structure. (**b**) Schematic diagrams of the structures of individual segmented proteins.

**Figure 2 vaccines-12-01205-f002:**
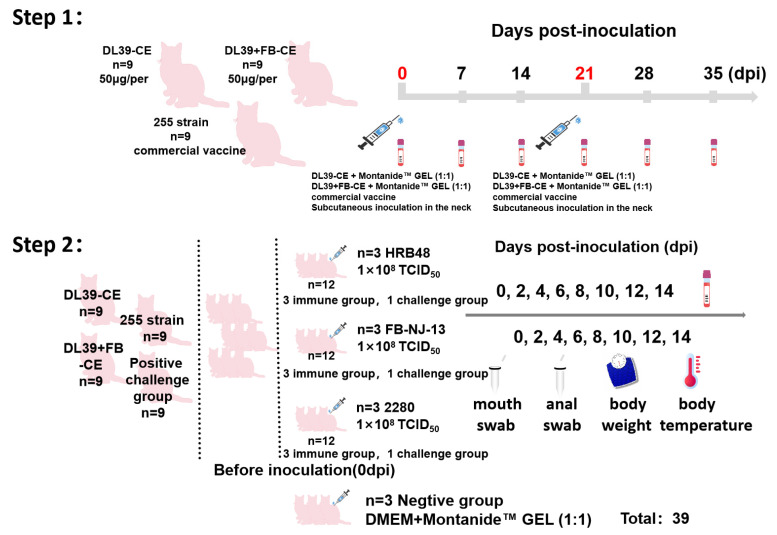
Schematic representation of the immunological attack of the CE_39_ and CE_39_-CE_FB_ recombinant proteins. The experiment was executed in a two-phase process. In the first phase, the cats were immunized with comprehensive immunization procedures meticulously annotated in the accompanying figure. Subsequently, in the second phase, the immunized cats were subjected to viral challenge, and the detailed challenge procedures were similarly meticulously annotated in the figure.

**Figure 3 vaccines-12-01205-f003:**
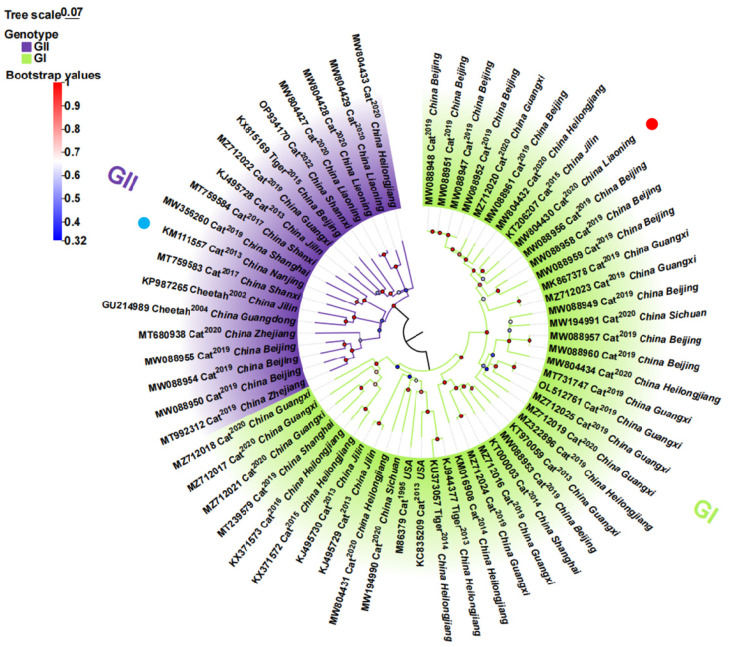
The ML phylogenetic tree estimated from the genome nucleotide sequences of FCV stains circulating in China as well as F9 and 2280. Note: Phylogenetic analysis was performed via MEGA-X software (10.1.8), and the model was JTT+G+I with 1000 replicates. Each genotype is shaded by a different color, and the DL39 strain and FB-NJ-13 strain are marked by a red dot and blue dot, respectively.

**Figure 4 vaccines-12-01205-f004:**
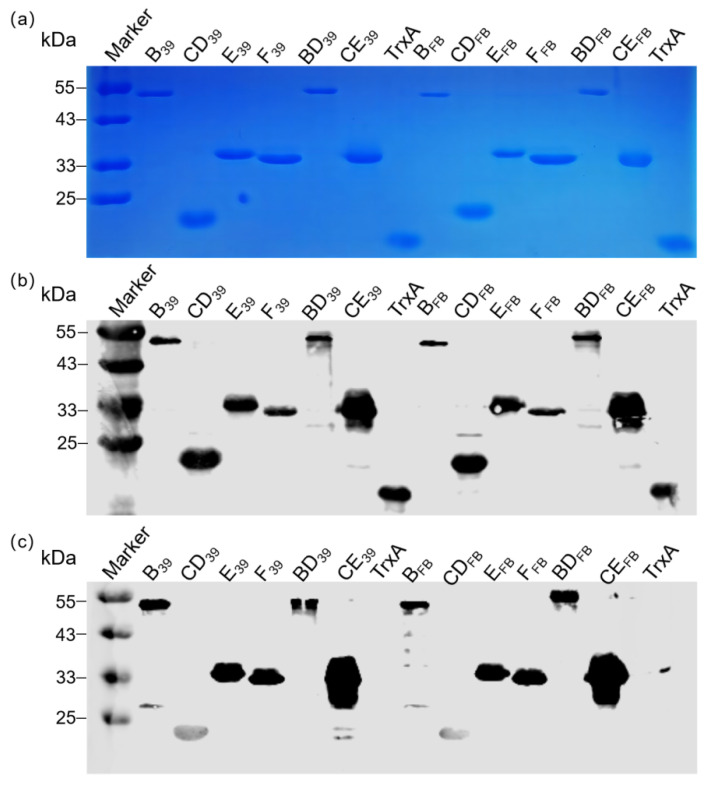
Schematic diagram and identification of purified recombinant proteins. (**a**) SDS–PAGE analysis of purified recombinant proteins. (**b**) Wb specificity identification results of purified recombinant proteins using commercial His mouse antibody as the primary antibody. (**c**) Wb specificity identification results of purified recombinant proteins using mouse anti-2280 serum as the primary antibody.

**Figure 5 vaccines-12-01205-f005:**
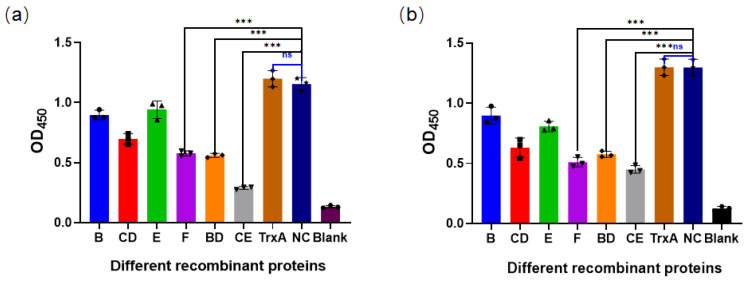
Results of competitive ELISA assays. (**a**) Competitive ELISA results of various recombinant proteins from DL39 against the parent strain. (**b**) Competitive ELISA results of various recombinant proteins from FB-NJ-13 against the parent strain. “ns” denotes no significant difference, and “*” denotes significant difference ( “***”, *p* < 0.001).

**Figure 6 vaccines-12-01205-f006:**
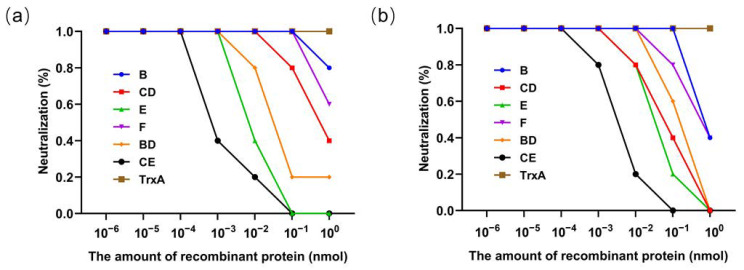
Results of competitive neutralization tests. (**a**) Results of the competitive neutralization test between the feline antiserum of DL39 and various recombinant proteins. (**b**) Results of the competitive neutralization test between the feline antiserum of FB-NJ-13 and various recombinant proteins.

**Figure 7 vaccines-12-01205-f007:**
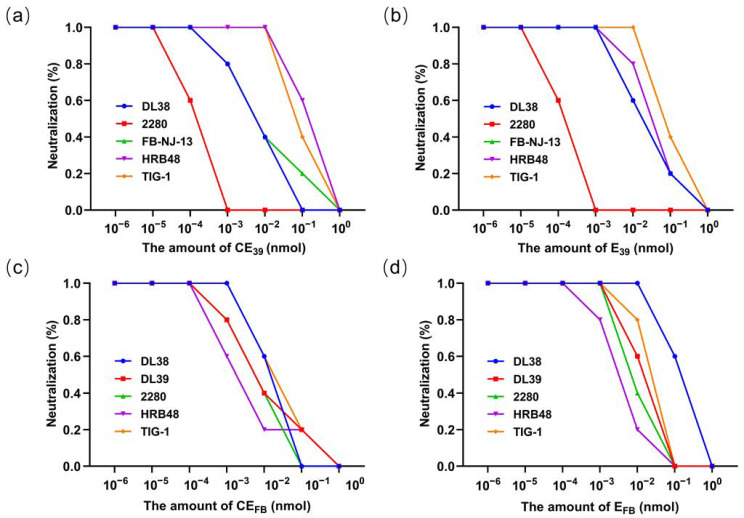
Results of competitive neutralization assays for the recombinant proteins CE and E. (**a**) Competitive neutralization assay result for the CE_39_ protein. (**b**) Competitive neutralization assay result for the E_39_ protein. (**c**) Competitive neutralization assay of the resulting CE_FB_ protein. (**d**) Competitive neutralization assay results for the E_FB_ protein.

**Figure 8 vaccines-12-01205-f008:**
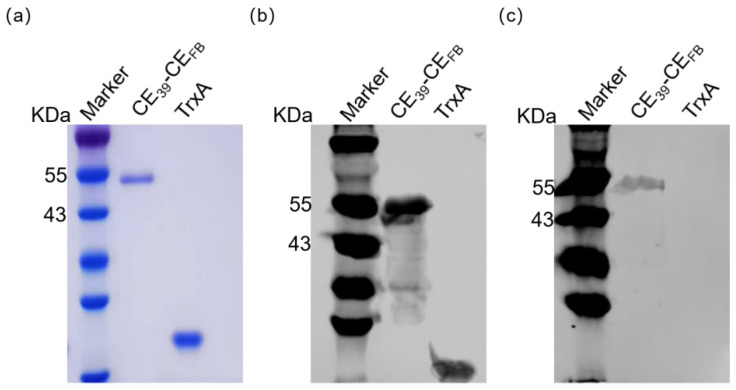
Identification of purified recombinant proteins. (**a**) SDS–PAGE analysis of purified CE_39_-CE_FB_ proteins. (**b**) Wb specificity identification results of purified CE_39_-CE_FB_ proteins using commercial His mouse antibody as the primary antibody. (**c**) Wb specificity identification results of purified CE_39_-CE_FB_ proteins using mouse anti-2280 sera as the primary antibody.

**Figure 9 vaccines-12-01205-f009:**
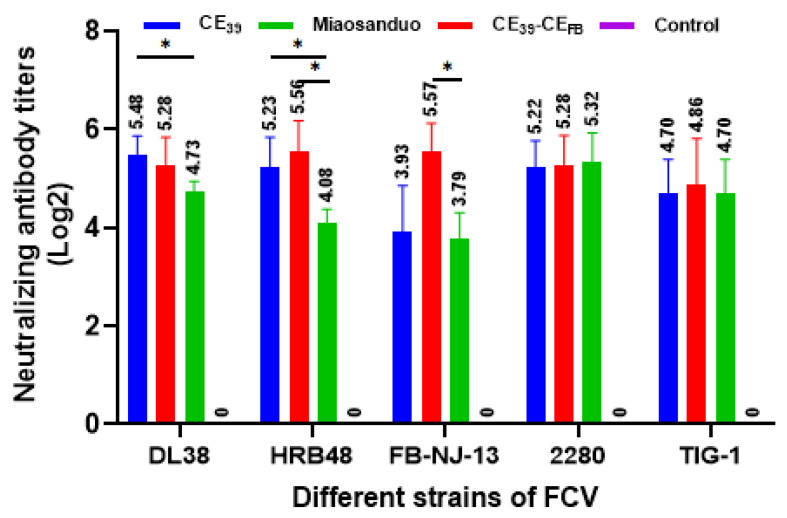
Results of the cross-neutralization test between feline anti-recombinant protein serum and the various strains. “*” denotes significant difference (*p* < 0.05).

**Figure 10 vaccines-12-01205-f010:**
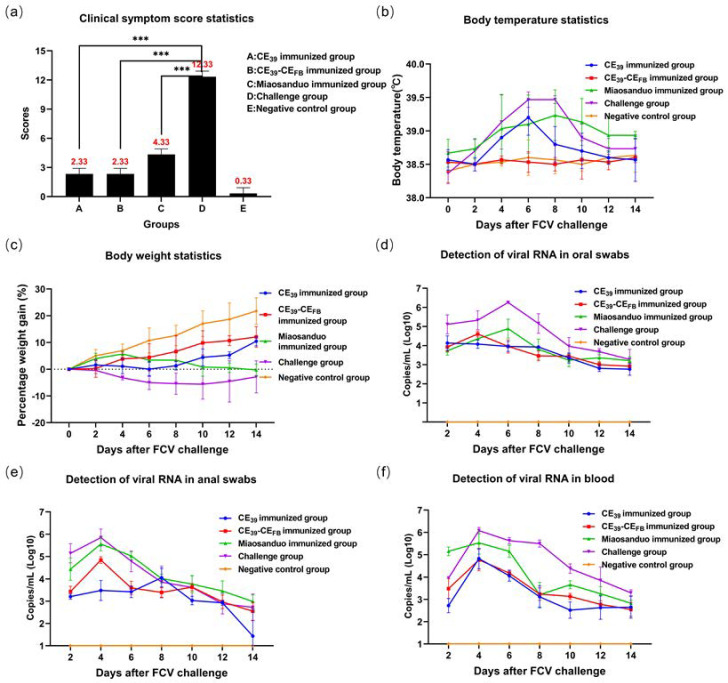
The results of the animal challenge test with the HRB48 strain. (**a**) Clinical symptom score of each group (“***” *p* < 0.001). (**b**) Body temperature measurement. (**c**) Body weight measurement. (**d**–**f**) Virus load measurements in oral swabs, anal swabs, and blood.

**Figure 11 vaccines-12-01205-f011:**
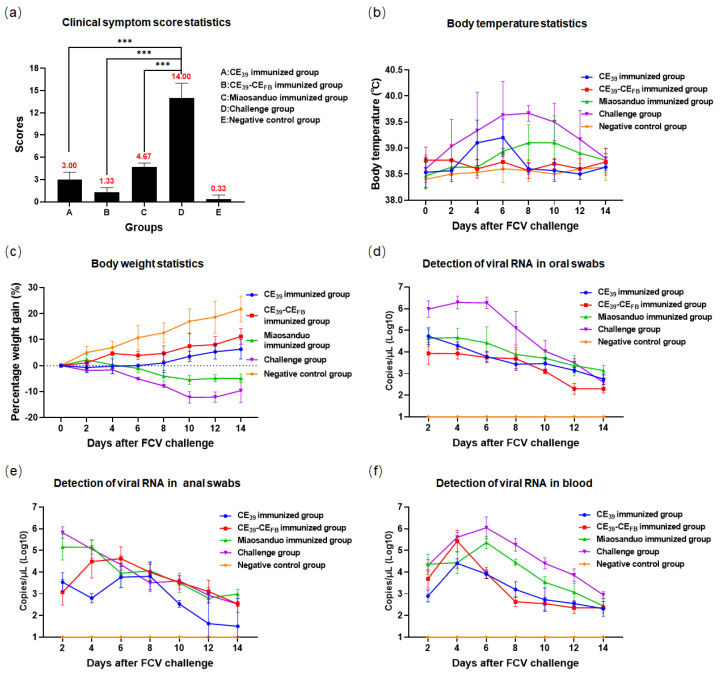
The results of the animal challenge test with FB-NJ-13 strain. (**a**) Clinical symptom score of each group (“***” *p* < 0.001). (**b**) Body temperature measurement. (**c**) Body weight measurement. (**d**–**f**) Virus load measurements in oral swabs, anal swabs, and blood.

**Figure 12 vaccines-12-01205-f012:**
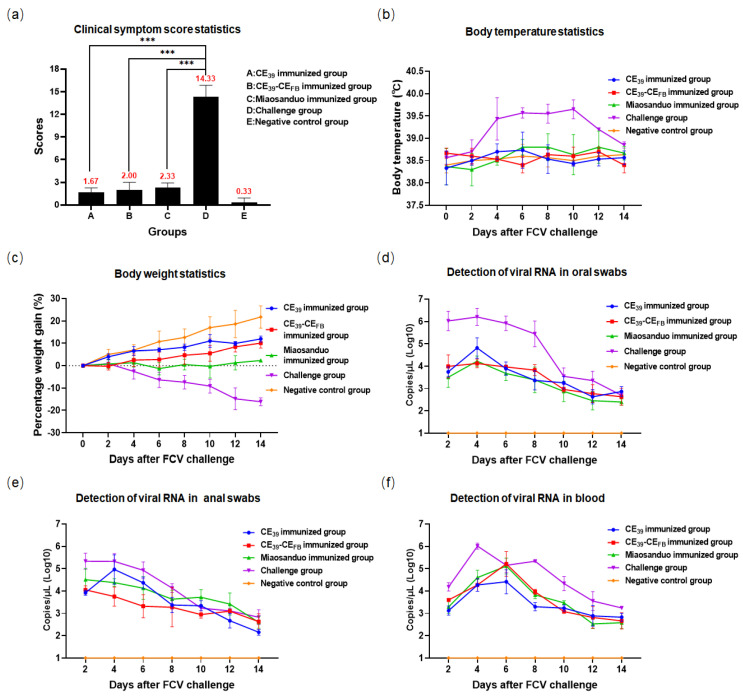
The results of the animal challenge test with 2280 strain. (**a**) Clinical symptom score of each group (“***” *p* < 0.001). (**b**) Body temperature measurement. (**c**) Body weight measurement. (**d**–**f**) Virus load measurements in oral swabs, anal swabs, and blood.

**Figure 13 vaccines-12-01205-f013:**
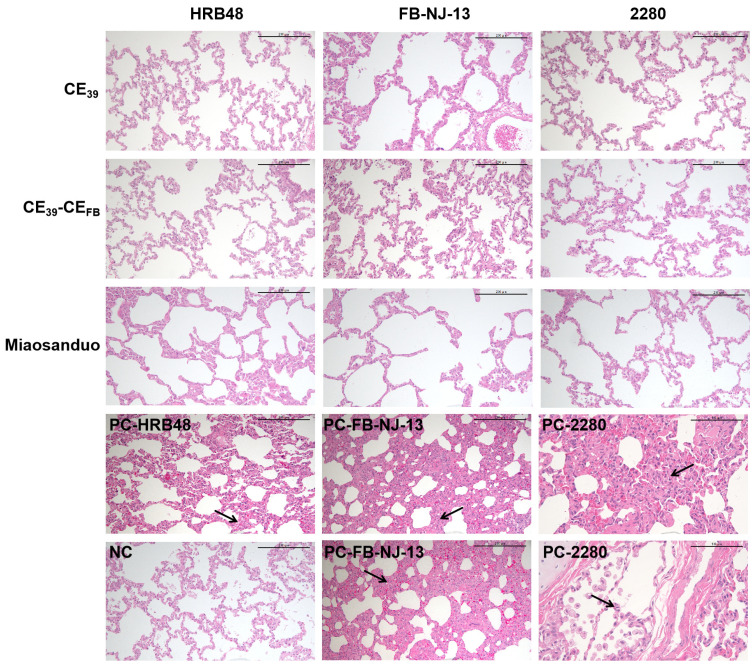
Histopathological images of lung tissue section (scale, 200 μM). Note: The name of the virus in the challenge group is indicated in the upper left corner, and the black arrow indicates the site of the tissue lesions.

## Data Availability

The raw data used for the preparation of figures in this study are available from the corresponding author upon reasonable request.

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
