# Peer review of "Screening and Immune Efficacy Evaluation of Antigens with Protection Against Feline Calicivirus"

_vaccines, 2024, doi:10.3390/vaccines12111205_

Round 1

Reviewer 1 Report

Comments and Suggestions for Authors

This paper reports an evaluation of the immunogenic potential of the different antigenic regions of FCV capsid protein expressed as recombinant proteins. The results indicate that a construct encompassing C, D and E antigenic regions (CE) rendered the highest immunogenicity affording protection against FCV challenges with different strains.

I think the study provides relevant results, although the manuscript has some issues that need to be addressed.

Comments:

- The manuscript does not have line numbers, which has hampered the review process.

- In this paper is quite important to precisely explain the recombinant proteins used in the study (i.e. the FCV antigenic regions B, C, D, E…). So there should be a figure in the Results section (not in Supplementary Materials), showing a schematic representation of the FCV capsid protein, its antigenic regions and the truncated products expressed (similar to Figure S1). However, Tables 1 and 2, specifying the optimal expression conditions or the yields of the purified recombinant proteins, are technical details not very relevant to follow the study, that can be shown in Supplementary Materials.

- It seems the recombinant proteins generated contain a “TrxA tag”, as indicated in the first paragraph of page 9 (“indicating that the TrxA tag attached to the recombinant proteins did not affect the experimental results”). However, this was not indicated previously in the corresponding Materials and Methods sections (2.3, 2.4) or in Results (3.2 section). For this reason, it is not understood why TrxA protein is included in the Western blots, ELISAs and neutralization assays performed with the recombinant proteins. Please correct the corresponding sections of Materials and Methods and Results to adequately describe the recombinant constructs expressed indicating all its components (i.e., TrxA, His-tag, FCV antigenic region, etc.).

- Regarding the phylogenetic analyses of the FCV genomic sequences, authors should indicate (in MM section 2.1, Results 3.1 and Figure 2) that the study was conducted only with strains circulating in China.

- Regarding the FCV clinical symptoms score used, it seems strange that the scores reached by the control groups for strains HRB48 and FB-NJ-13 (i.e. normal “mild” FCV strains): 12.33 and 14.00 respectively, were very similar to that obtained using the virulent systemic strain 2280 (score 14.33), when this latter strain should be quite more virulent, thus causing more severe symptoms, even inducing the death of some infected cats (one of the three cats from the positive control group died). Maybe the clinical symptoms score used (Table S3) may need to be revised. For example, it does not include “lethargy”, which according to the text affected several 2280-infected cats, but it is not mentioned as a symptom induced by the other FCV strains.

- When describing the immunogenicity effect of the CE-based FCV recombinant proteins, the authors state the constructs “conferred 100% protection rate against FCV challenge”, with the three strains used in the study. However, in all three cases they observed FCV-related symptoms in the CE-immunized groups (i.e., depression, fever, weight loss, ocular secretions, lethargy, mild respiratory distress…). The data shown in figures 9-11 indicate the CE-immunized groups exhibited less clinical symptoms than the challenge control groups, or even the groups immunized with the commercial vaccine, Ok, but it is not clear the results obtained can be regarded as “100% protection rate”.

- Throughout the paper some references cited are not related with the statement made above. For example:

            - Page 2: “The ORF2 gene encodes the capsid protein VP1, which is divided into six regions, A to F, and serves as the primary antigenic protein of FCV [7]” (This reference deals about VP2, which is ORF3).

- Page 2: “The single serotype of FCV may be associated with the E region of the VP1 protein [8,9]” (This references have no relation with VP1 antigenic regions or serotypes).

- Page 2: “Currently, worldwide reports of FCV outbreaks have gradually increased, indicating the widespread prevalence of the virus globally [17,18]” (again, the references cited are not related with the statement).

Minor Comments

- In Abstract: “leading to a singular prevention and control method”. I think by “singular” the authors mean “single”.

- In Introduction at the end of the first paragraph: “biotin labelling experiments” I think it is better: Pepscan (or peptide library) epitope mapping.

- In Introduction at the second paragraph: “this disease has rapidly spread and become prevalent in multiple countries [22-27]”. The FCV virulent systemic disease (VSD-FCV) has been reported in several countries, yes, but has it become “prevalent”? Are the authors sure about this? I think this is not what the cited references indicate.

- In section 2.2, Awkward sentence: “which was shown previously demonstrated to possess”, please correct.

- In section 2.8: “Group 14” is wrong. Should be Group 13.

- In section 3.1: “(18/61)” is wrong. Should be: (19/61).

-Figure 1 “commercialization” I guess it is commercial vaccine.

- Page 17 first paragraph: “Research has revealed that the hypervariable region at the 5' end and the C region of the E domain” This sentence is confusing. What they call “C region of the E domain” is the “conserved central part of antigenic region E” (Econsv in figure S1a), which was not previously described in the manuscript, and it is not related to the antigenic region “C” cited throughout the paper (as seen in figure S1a, that’s why this figure should be incorporated to the manuscript). By the way: B, C, D, E, F are “antigenic regions”, not “domains”, this should be corrected throughout the text.

Author Response

I deeply appreciate the time and effort you have dedicated from your hectic schedule to offer valuable guidance on my paper. We have diligently responded to each of your suggestions and incorporated the necessary corrections into the manuscript. Your insightful advice has played a pivotal role in improving the fluency, coherence, and rigorousness of our paper. I extend my heartfelt best wishes to you. My specific answers as follows:

Comments and Suggestions for Authors

This paper reports an evaluation of the immunogenic potential of the different antigenic regions of FCV capsid protein expressed as recombinant proteins. The results indicate that a construct encompassing C, D and E antigenic regions (CE) rendered the highest immunogenicity affording protection against FCV challenges with different strains.

I think the study provides relevant results, although the manuscript has some issues that need to be addressed.

Comments:

1.- The manuscript does not have line numbers, which has hampered the review process.

Response: Thank you for pointing this out. We agree with this comment. We have re-annotated the line numbers in the manuscript to facilitate your review process.

2.- In this paper is quite important to precisely explain the recombinant proteins used in the study (i.e. the FCV antigenic regions B, C, D, E…). So there should be a figure in the Results section (not in Supplementary Materials), showing a schematic representation of the FCV capsid protein, its antigenic regions and the truncated products expressed (similar to Figure S1). However, Tables 1 and 2, specifying the optimal expression conditions or the yields of the purified recombinant proteins, are technical details not very relevant to follow the study, that can be shown in Supplementary Materials.

Response: Thank you for pointing this out. We agree with this comment. Therefore, we have annotated the antigenic regions such as B, CD, E, and F in the predicted structural diagram of the VP1 protein and combined them with Figure S1 to create a new image as Figure 1. Additionally, we have moved Table 1 and Table 2 to the supplementary materials. Your suggestions have greatly enhanced the clarity of our article's ideas and rigor of its structure. Lines 116-118.

3.- It seems the recombinant proteins generated contain a “TrxA tag”, as indicated in the first paragraph of page 9 (“indicating that the TrxA tag attached to the recombinant proteins did not affect the experimental results”). However, this was not indicated previously in the corresponding Materials and Methods sections (2.3, 2.4) or in Results (3.2 section). For this reason, it is not understood why TrxA protein is included in the Western blots, ELISAs and neutralization assays performed with the recombinant proteins. Please correct the corresponding sections of Materials and Methods and Results to adequately describe the recombinant constructs expressed indicating all its components (i.e., TrxA, His-tag, FCV antigenic region, etc.).

Response: Thank you for pointing this out. Indeed, our description in the text regarding this aspect is unclear. We utilized the pET-32a vector, which harbors both a TrxA tag and a His tag. The TrxA tag serves to enhance protein expression, while the His tag facilitates subsequent identification of the protein. Consequently, each expressed protein consists of a TrxA tag, the antigenic region, and a His tag. In subsequent experiments, we verified that the TrxA tag had no impact on our tests; therefore, we retained this tag, as tag removal can cause some degree of protein loss. Your suggestions are invaluable to us, and we have made corrections in Sections 2.3, 2.4, and 3.2 accordingly. Lines 112, 128.

4.- Regarding the phylogenetic analyses of the FCV genomic sequences, authors should indicate (in MM section 2.1, Results 3.1 and Figure 2) that the study was conducted only with strains circulating in China.

Response: Thank you for pointing this out. We agree with this comment. We have made corrections in Section 2.1, Section 3.1, and Figure 2 of the manuscript (lines 92, 264, 278).

5.- Regarding the FCV clinical symptoms score used, it seems strange that the scores reached by the control groups for strains HRB48 and FB-NJ-13 (i.e. normal “mild” FCV strains): 12.33 and 14.00 respectively, were very similar to that obtained using the virulent systemic strain 2280 (score 14.33), when this latter strain should be quite more virulent, thus causing more severe symptoms, even inducing the death of some infected cats (one of the three cats from the positive control group died). Maybe the clinical symptoms score used (Table S3) may need to be revised. For example, it does not include “lethargy”, which according to the text affected several 2280-infected cats, but it is not mentioned as a symptom induced by the other FCV strains.

Response: Thank you for pointing this out. The scoring of clinical symptoms associated with feline calicivirus (FCV) infection has consistently garnered significant attention, and we have been adhering to the established scoring criteria.  Your suggestions are indeed rational;  however, in our practical setting, the most prevalent clinical manifestations of FCV encompass oral ulcerations, weight reduction, elevated body temperature, and cough etc, whereas mortality is relatively uncommon.  Consequently, our scoring criteria prioritize the emphasis on typical clinical symptoms, with symptoms such as lethargy being subsumed within other symptoms.  Additionally, although the FB-NJ-13 strain does not constitute a virulent systemic disease (VSD) strain, its pathogenicity remains potent, capable of eliciting rather severe clinical manifestations. Notably, in our previous animal models, no mortality was observed, thereby precluding its classification as a VSD strain. In summary, it is plausible that the clinical symptom scores of the FB-NJ-1 strain and the 2280 strain are similar. In conclusion, we aim to negotiate with you to retain the current scoring system, as it holds immense significance for our animal experiments involving prevalent strains. Nevertheless, we will continue to monitor changes in the pathogenicity of prevalent strains, and if numerous novel clinical symptoms emerge, we will strive to update our scoring criteria accordingly.

6.- When describing the immunogenicity effect of the CE-based FCV recombinant proteins, the authors state the constructs “conferred 100% protection rate against FCV challenge”, with the three strains used in the study. However, in all three cases they observed FCV-related symptoms in the CE-immunized groups (i.e., depression, fever, weight loss, ocular secretions, lethargy, mild respiratory distress…). The data shown in figures 9-11 indicate the CE-immunized groups exhibited less clinical symptoms than the challenge control groups, or even the groups immunized with the commercial vaccine, Ok, but it is not clear the results obtained can be regarded as “100% protection rate”.

Response: Thank you for pointing this out. We agree with this comment. In our conclusions, we have removed the phrase “100% protection rate,” as this approach is indeed controversial. As you have pointed out, although the subunit vaccine demonstrates better efficacy than the currently marketed vaccine strain FCV-255, some clinical symptoms still manifest. Furthermore, our validation is limited to preliminary animal trials, rendering this conclusion inaccurate. However, our immunogroup exhibited milder clinical symptoms, reduced virus excretion, and an absence of oral ulcerations. In the subsequent stages, we will continue to refine the evaluation of the vaccine. Your suggestion Is Invaluable for enhancing the rigor and accuracy of our manuscript. Thank you for your precious feedback. Specific modification location: line 430, 460, 480, 631, 635.

7.- Throughout the paper some references cited are not related with the statement made above. For example:

- Page 2: “The ORF2 gene encodes the capsid protein VP1, which is divided into six regions, A to F, and serves as the primary antigenic protein of FCV [7]” (This reference deals about VP2, which is ORF3).

- Page 2: “The single serotype of FCV may be associated with the E region of the VP1 protein [8,9]” (The references have no relation with VP1 antigenic regions or serotypes).

- Page 2: “Currently, worldwide reports of FCV outbreaks have gradually increased, indicating the widespread prevalence of the virus globally [17,18]” (again, the references cited are not related with the statement).

Response: Thank you for pointing this out. We agree with this comment. We have addressed the concerns you raised regarding the citation of references and have conducted a review of the other references entries (lines 47-49, 59).

Minor Comments

1.- In Abstract: “leading to a singular prevention and control method”. I think by “singular” the authors mean “single”.

Response: Thank you for pointing this out. We agree with this comment. Revisions have been made in the manuscript (line 14).

2.- In Introduction at the end of the first paragraph: “biotin labelling experiments” I think it is better: Pepscan (or peptide library) epitope mapping.

Response: Thank you for pointing this out. We agree with this comment. Revisions have been made in the manuscript (line 54).

3.- In Introduction at the second paragraph: “this disease has rapidly spread and become prevalent in multiple countries [22-27]”. The FCV virulent systemic disease (VSD-FCV) has been reported in several countries, yes, but has it become “prevalent”? Are the authors sure about this? I think this is not what the cited references indicate.

Response: Thank you for pointing this out. We agree with this comment. We have replaced “prevalent” with “reported,” which renders the description more accurate. Revisions have been made in the manuscript (line 63).

4.- In section 2.2, Awkward sentence: “which was shown previously demonstrated to possess”, please correct.

Response: Thank you for pointing this out. We agree with this comment. Revisions have been made in the manuscript (line 102-103).

5.- In section 2.8: “Group 14” is wrong. Should be Group 13.

Response: Thank you for pointing this out. We agree with this comment. Revisions have been made in the manuscript (line 223).

6.- In section 3.1: “(18/61)” is wrong. Should be: (19/61).

Response: Thank you for pointing this out. We agree with this comment. Revisions have been made in the manuscript (line 269).

7.-Figure 1 “commercialization” I guess it is commercial vaccine.

Response: Thank you for pointing this out. We agree with this comment. Revisions have been made in the manuscript (Figure 1).

8.- Page 17 first paragraph: “Research has revealed that the hypervariable region at the 5' end and the C region of the E domain” This sentence is confusing. What they call “C region of the E domain” is the “conserved central part of antigenic region E” (Econsv in figure S1a), which was not previously described in the manuscript, and it is not related to the antigenic region “C” cited throughout the paper (as seen in figure S1a, that’s why this figure should be incorporated to the manuscript). By the way: B, C, D, E, F are “antigenic regions”, not “domains”, this should be corrected throughout the text.

Response: Thank you for pointing this out. We agree with this comment. Firstly, we have revised Figure S1 to Figure 1 and incorporated structural diagrams of various antigens. Additionally, we have modified the descriptions of antigens in different regions within the text, changing "domains" to "antigenic regions". This was indeed an obvious error, and we appreciate your suggestion. Meanwhile, we have also revised sentences containing grammatical errors. Specific modification location: lines 510-514.

Reviewer 2 Report

Comments and Suggestions for Authors

A great paper covering a lot of both in vivo and in vitro work on an important topic for feline (and other calicivirus) vaccinology. My comments are many, but all are relatively minor. 

The two main ones relate to the quality of the discussion, and that perhaps there is an opportunity to relate this work to that going on for other human and animal caliciviruses.

Title – I don’t think epidemic is the right word. Nor do I believe you can justify “broad-spectrum protection” in the title.  Perhaps the title can be shortened and revised.

17 – instead of “revealing” perhaps “confirming”?

Line 29 – “protected cats”. Perhaps that is not quite right. Reduced clinical signs… but vaccinated cats still became infected.

47 – major capsid protein…. has been divided into

47 – the sentence starting “the single serotype…. “ . I am neither sure what this means not of the relevance of the two references.

50 – “amino acid mutation” is perhaps better… all bases are likely to mutate, just some are selected.

48-56 – this is an important section but to me seems jumbled and overly complex… and perhaps in the wrong order.

60 – I don’t think anyone has described numerous genotypes. Reference or remove. Nor a gradual increase in reports. The references for this are also inappropriate.

64 – do you believe this? Or is it possible just more people are looking now. I think it is important we don’t accidentally mislead the reader.

67 – again, what do you mean by “prevalent”. People reading this may think there are frequent outbreaks all over the world of fatal disease. My impression is that is not the case. Giving this 6 references is probably excessive.

70- vaccines do nothing to reduce prevalence of FCV. To reduce the incidence and severity of disease?

96 strains of FCV?

93-96 – just say genbank accession number for the first one, it is obvious thereafter.

97 – preserved? Probably needs more detail. How were they made? How many cats? Infection? Challenge? Etc

101 – these sequences were….

103 “necessary” is a strange work here. If the features were described in ref 39 then just say “as previously described”

108 – I would delete “optimal protective”

110 demonstrated … and good.

114 – probably should also say what genotype is DL39

118 – how was correct amplification confirmed?

118/9 – was anything done to check the sequence of the constructed plasmids?

129 – how was concentration determined?

147 – which company at what dilution?

134 – again we need to know more about how these antisera were made… and perhaps their homologous titre?

167 – instead of “parental” you might say “homologous”

169 – how much of the homologous virus was added?

174  - how did you determine “damaged cells”?

182 – how were these strains chosen? And note here you use the word “strain” whereas in the introduction you don’t.

This reviewer would find it helpful if the competitive assays were introduced by a brief description of their logic. If I get it right, antigenic proteins will reduce the neutralisation of the specific antisera? Interestingly you do this yourself on line 308-309. Please bring to the methods as well to make it easier to follow.

188 – briefly how did you determine the optimal conditions? Even “standard protocols” would help.

Figure 1 – sorry this needs a more detailed legend.

194 – the selection of challenge strains…

195 - the original host?

196 – I don’t think choosing three strains can qualify as being “representative” of the diversity so beautifully shown in figure 2. This is a fundamental challenge with FCV, choosing valid challenges to test cross-reactivity. Perhaps this could be included as a limitation or an area for future research.

204 – a strange assay to use to confirm lack of exposure. I guess ok. But would it be better to look for neutralising antibodies to the challenge strains? Perhaps consider if you can defend the next line “negative to FCV antibodies”

208 – challenge

202 – says 39. 3 x 14 = 42.

210-216 is repetitive with the preceding section.

218 says “second” – but that is not described above.

230 – I guess you should say how you quantified the virus

2.8 challenge test – were any of the observers blinded to the groups the cats were in?

238 – add that these are obtained from China.

239 – phylogenetic analysis does not evidence prevalence. It does talk of diversity.

Line 240 is repetitive with what follows

244 – there are no leopard sequences in the tree.

254 – the red and blue dot do not help…. and are not needed – they are easy to see.

260 – personally I think these data including table 1 are better suited to the methods

Figure 2 – are the two strains used in your experiments also in this phylogeny. Assuming they are they should be highlighted clearly. To simplify the figure consider removing “china” from each name.

Line 261-262 are not needed. Indeed you could probably start this section around line 262.

Line 264-5 can go – it is methods

Figure 3 – are all three panels needed?

Table 2 could also be removed – you could just say concentrations ranged from x-y.

295 – reactogenicity is not a word I know. Sorry.

Section 3.3 – this can be simplified by dealing with both strains together. In both caess, CE had the most profoud effect. F and BD were also significant.

311 and elsewhere (eg line 321) – I am not sure it is right to describe these antigens as having “neutralising activity”.

358 – perhaps remind us of the strain that Miaosanduo is based on?

359 – if comparing probably should add a statistical test for significance?

363 – I am not sure you can call them “epidemic” strains.

369 – all challenged groups – I think you mean those not vaccinated?

378 – figure 9b,c does not talk about ulcers

379 – they are nearly all challenge groups. Use the group names you defined earlier, perhaps put in brackets (control)

230 – describe method of quantification of virus

378 – you have not measured virus shedding as this is a genome-based assay. In my opinion, a simple virus isolation may have been more informative but not necessary.

Figure 9a – could this be coloured the same as the rest of the figure? And please explain in the methods probably what you have done to the recorded clinical scores to produce this overall score.

448 – you have two control groups. Non-challenge control group. Again use the group designations you defined earlier.

411 – delete “extremely high”

414 – 415 – not sure you can defend “100% protection”. Remove this last sentence

What has happened to the results of the HRB48 challenge.

452 – challenged groups is again misleading. I think you mean unvaccinated controls.

Figure 12 legend needs improving – the bottom six panels I think are unvaccinated controls. It might be better if the column was the challenge stain, and the row whether or not the cat was vaccinated. The arrows do not help. The entire view shows abnormal pathology (although I am not a pathologist). Use the arrows to point to more specific things described in the text.

475 – domestic Chinese

477-479 – I sense trivalent is misleading here. I don’t think there is evidence that vaccinated cats are at increased risk is there? And vaccines are not licensed to protect chronic stomatitis. I guess it depends what you mean by “stomatitis”. Think this sentence needs careful thought. 

527 – 535 – can be removed. Or just briefly mentioned as future work. Your experiments did not (sorry if I missed it) include a non-adjuvanted control so you don’t even know if the addition of an adjuvant had an effect. Perhaps this should Be mentioned as a limitation.

Line 543 onwards is conclusion. This can be considerably shortened.

Discussion – for me the discussion lacks focus and is disappointing. Interesting things you could have considered – is it usual for vaccinated cats to shed post-challenge, did the 39/fb recombinant improve protection, what about a particular strain makes it broadly cross-reactive, could the conserved regions in the peptides be contributing to cross-reactivity, do you feel your peptides contain relevant conformation, or are they more likely to be linear. Perhaps their results could be compared to other caliciviruses like human norovirus or RHDV where I suspect this approach has been more considered. Indeed there is a recombinant myxo:RHDV vaccine on the market and as for the human noroviruses,  culturing is not an option. Adding this human dimension would also broaden the interest of this paper. Indeed it might be worth stating in the introduction that this antigen approach has been tried for a range of other viruses especially human norovirus.  I am sure the authors can think of other discussion points. The large section on adjuvants seems largely irrelevant.

Author Response

I deeply appreciate the time and effort you have dedicated from your hectic schedule to offer valuable guidance on my paper. We have diligently responded to each of your suggestions and incorporated the necessary corrections into the manuscript. Your insightful advice has played a pivotal role in improving the fluency, coherence, and rigorousness of our paper. I extend my heartfelt best wishes to you. My specific answers as follows:

Comments and Suggestions for Authors

A great paper covering a lot of both in vivo and in vitro work on an important topic for feline (and other calicivirus) vaccinology. My comments are many, but all are relatively minor.

The two main ones relate to the quality of the discussion, and that perhaps there is an opportunity to relate this work to that going on for other human and animal caliciviruses.

Response: Thank you for pointing this out. We have already added in discussion.

Title – I don’t think epidemic is the right word. Nor do I believe you can justify “broad-spectrum protection” in the title.  Perhaps the title can be shortened and revised.

 Response: Thank you for pointing this out. We agree with this comment. We have revised the title to “Screening and Immune Efficacy Evaluation of Antigens with Protection Against Feline Calicivirus”. Lines:2-3.

17 – instead of “revealing” perhaps “confirming”?

Response: Thank you for pointing this out. We agree with this comment. Revisions have been made in the manuscript (line 16).

Line 29 – “protected cats”. Perhaps that is not quite right. Reduced clinical signs… but vaccinated cats still became infected.

Response: Thank you for pointing this out. We agree with this comment. Revisions have been made in the manuscript (lines 28-29).

47 – major capsid protein…. has been divided into

Response: Thank you for pointing this out. We agree with this comment. Revisions have been made in the manuscript (line 46).

47 – the sentence starting “the single serotype…. “ . I am neither sure what this means not of the relevance of the two references.

Response: Thank you for pointing this out. We have deleted the sentence due to its inaccuracy and abruptness. Line 48.

50 – “amino acid mutation” is perhaps better… all bases are likely to mutate, just some are selected.

Response: Thank you for pointing this out. We agree with this comment. Revisions have been made in the manuscript (line 49).

48-56 – this is an important section but to me seems jumbled and overly complex… and perhaps in the wrong order.

Response: Thank you for pointing this out. We agree with this comment. We have moved the last sentence to the forefront, first introducing regions C, D, and E as the primary antigenic regions, and then progressively emphasizing region E, thereby enhancing the logical flow of the manuscript (line 47-54).

60 – I don’t think anyone has described numerous genotypes. Reference or remove. Nor a gradual increase in reports. The references for this are also inappropriate.

Response: Thank you for pointing this out. We agree with this comment. The term "numerous" is incorrect, and we replace it with "different". Additionally, the assertion that FCV is widespread globally is inaccurate, and therefore, we have also made revisions to this point in the manuscript. The references cited are articles that possess certain reference significance. Lines 56-58.

64 – do you believe this? Or is it possible just more people are looking now. I think it is important we don’t accidentally mislead the reader.

Response: Thank you for pointing this out. We agree with this comment. The previous sentence incorporated our own predictions, which were inaccurate, and therefore we have revised it accordingly. The modifications are as follows: In China, multiple regions have reported incidents of FCV infecting feline species in recent years. Lines 58-60.

67 – again, what do you mean by “prevalent”. People reading this may think there are frequent outbreaks all over the world of fatal disease. My impression is that is not the case. Giving this 6 references is probably excessive.

Response: Thank you for pointing this out. Indeed, our previous statement was inaccurate, and we have accordingly made modifications in the manuscript. Furthermore, we have revised the terminology, including the words "transmission" and "epidemic," to ensure accuracy. VSD-FCV has been reported multiple times globally, and our intention in citing those references is to provide readers with a better understanding of the characteristics and epidemiological trends of the disease. Therefore, we hope that the cited references can be retained. Lines 61-63.

70- vaccines do nothing to reduce prevalence of FCV. To reduce the incidence and severity of disease?

Response: Thank you for pointing this out. We agree with this comment. Revisions have been made in the manuscript (line 66).

96 strains of FCV?

Response: Yes, the revisions have been made in the manuscript (line 87).

93-96 – just say genbank accession number for the first one, it is obvious thereafter.

Response: Thank you for pointing this out. We have removed the unnecessary information as per your suggestion. Revisions have been made in the manuscript (lines 85-87).

97 – preserved? Probably needs more detail. How were they made? How many cats? Infection? Challenge? Etc

Response: The antisera utilized in this study were previously prepared cat antisera, which were generated by immunizing cats with inactivated viruses. Three cats were included in each group, and sera were collected intravenously after the second immunization. However, since these antisera belong to the public materials of our laboratory, we have merely indicated that they were sera stored within our laboratory.

101 – these sequences were….

Response: Thank you for pointing this out. The specific details of these sequences, which primarily consist of 61 domestic virus strains along with F9 and FCV 2280, have been delineated within the evolutionary tree. Line 91.

103 “necessary” is a strange work here. If the features were described in ref 39 then just say “as previously described”

Response: Thank you for pointing this out. We agree with this comment. Revisions have been made in the manuscript (lines 94-95).

108 – I would delete “optimal protective”

Response: Thank you for pointing this out. We agree with this comment. We deleted the “optimal protective”. Revisions have been made in the manuscript (line 100).

110 demonstrated … and good.

 Response: Thank you for pointing this out. We also found that the description in that sentence was inaccurate, and thus we made revisions in the manuscript. The modified content is as follows:which could provide broad-spectrum protection according to our previous study. Line 102.

114 – probably should also say what genotype is DL39

Response: Thank you for pointing this out. We agree with this comment. We incorporated the genotype GI following the DL39 strain. Revisions have been made in the manuscript (line 102).

118 – how was correct amplification confirmed?

Response: Thank you for pointing this out. After amplification, we conducted sequencing of the fragments to ensure the accuracy of the sequence results. In the manuscript, we have provided supplementary descriptions. Line 110-111.

118/9 – was anything done to check the sequence of the constructed plasmids?

Response: Thank you for pointing this out. After constructing the plasmid, we need to sequence the constructed plasmid to confirm its accuracy. In the manuscript, we have provided supplementary descriptions. Lines 110-111.

129 – how was concentration determined?

Response: Thank you for pointing this out. The concentration of the purified proteins was then determined with BCA Protein Assay Kit of Beiotime Biotechnology. In the manuscript, we have provided supplementary descriptions. Line 129.

147 – which company at what dilution?

Response: The question you posed has not been fully grasped by us, but we have made an attempt to provide an answer based on our understanding. All experiments conducted in this study were completed by members of our research group. For the dilution of positive sera, we utilized a ratio of 1:100 (diluent is PBS), which was previously determined as the optimal concentration for screening in our established FCV ELISA method. For the dilution of proteins, we employed quantitative dilution based on the molar mass of the proteins (0.1 nmol). Given that the size of each protein varies, it was necessary to select a method that allowed for consistent and accurate quantification of the proteins. Using the amount of substance (moles) for quantification was deemed an appropriate approach, as also reported in some literature. After diluting the serum and protein, 100 microliters of each were mixed and placed in a new ELISA plate for reaction. Upon completion of the reaction, the mixture was added to the ELISA reaction plate coated with antigen for further experimentation. Detailed procedures and results have been outlined in the manuscript. Line 155.

134 – again we need to know more about how these antisera were made… and perhaps their homologous titre?

Response: This serum, which has been preserved in our laboratory, was prepared following a procedure briefly described in the question pertaining to line number 97. The neutralization titer with homologous serum is presented in Figure S3 (steps of neutralization test). Therefore, it has not been elaborated in the ELISA experiment.

167 – instead of “parental” you might say “homologous”

Response: Thank you for pointing this out. We agree with this comment.  We revised the“parental” to “homologous”, and also made to other parts of the manuscript. Revisions have been made in the manuscript (line 135, 163, 177, 179, 306, 327, 337).

169 – how much of the homologous virus was added?

Response: The homologous virus was added in 100TCID50/50 μL. Please note: 100 TCID50 of fully diluted homologous virus, rather than a specified volume, were added to each well.

174  - how did you determine “damaged cells”?

Response: The pathological changes in FCV are highly apparent, and can be readily discerned through microscopic examination. To enhance the rigor of our experimental results, we classified all wells exhibiting any pathological changes as positive wells. This approach, to a certain extent, compensates for the errors associated with visual inspection by the naked eye.

182 – how were these strains chosen? And note here you use the word “strain” whereas in the introduction you don’t.

Response: In this experiment, the virus strains utilized primarily included HRB48, TIG-1, DL38, and 2280, in addition to two homologous strains. Efforts were made to select representative and prevalent virus strains. Specifically, HRB48 belongs to the GI genotype, TIG is a VSD strain isolated from tigers, DL38 is a GII genotype strain, and 2280 is a recognized virulent strain. Detailed information about these strains is indicated in Figure 2 of the evolutionary tree, and thus is not repeated here. In the Introduction section, we mainly introduced basic information about FCV without delving into specific virus strains, hence the term "strain" was not used.

This reviewer would find it helpful if the competitive assays were introduced by a brief description of their logic. If I get it right, antigenic proteins will reduce the neutralisation of the specific antisera? Interestingly you do this yourself on line 308-309. Please bring to the methods as well to make it easier to follow.

Response: Thank you for pointing this out. We have supplemented the manuscript with the principle of the competitive neutralization test. The principle of the competitive neutralization test is based on antigen competition. Initially, the protein is co-incubated with positive serum. If this protein binds to the neutralizing antibodies in the serum, the subsequent binding of the serum to the virus will result in a reduction of the neutralization titer. Lines 166-171.

188 – briefly how did you determine the optimal conditions? Even “standard protocols” would help.

Response: Thank you for pointing this out. We have incorporated the standard description into the manuscript. This is a particularly good suggestion. Thank you again. Lines 188-191.

Figure 1 – sorry this needs a more detailed legend.

Response: We have supplemented Figure 1 to include information on the immunization method, the number of experimental animals, and other relevant details. Furthermore, we have provided more specific descriptions for the figure annotations. Lines 211-216.

194 – the selection of challenge strains…

Response: For the selection of challenge viruses, we chose the prevalent strains HRB48 of GI genotype, FB-NJ-13 of GII genotype, and the internationally standardized virulent strain 2280. Among them, animal infection models have been successfully established in our laboratory for both HRB48 and FB-NJ-13 strains. Therefore, our challenge、 experiments utilized viruses of these two genotypes as well as the VSD strain.

195 - the original host?

Response: Our cats were uniformly purchased from a feline experimental base, with complete animal ethics approval numbers. From transportation to the conclusion of the experiments, we have consistently adhered to animal welfare standards for experimentation.

196 – I don’t think choosing three strains can qualify as being “representative” of the diversity so beautifully shown in figure 2. This is a fundamental challenge with FCV, choosing valid challenges to test cross-reactivity. Perhaps this could be included as a limitation or an area for future research.

Response: We agree with your perspective. Due to the high variability of FCV, its prevention and control have been challenging. When developing vaccines, it is impractical to test against all strains, so we rely on a combination of in vitro neutralization tests and in vivo challenge experiments for validation. For the in vitro neutralization tests, we screened six strains. For the in vivo challenge experiments, we selected strains with different virulence levels and genotypes as much as possible. To some extent, this already demonstrates the broad-spectrum nature of our approach. The issue you raised is indeed one of the most serious constraints on the efficacy of current FCV vaccines.

204 – a strange assay to use to confirm lack of exposure. I guess ok. But would it be better to look for neutralising antibodies to the challenge strains? Perhaps consider if you can defend the next line “negative to FCV antibodies”

Response: Using the method described in the manuscript to detect FCV antibodies in animals is a widespread and accepted approach. While detecting neutralizing antibodies against challenge strains is a more accurate but more difficult method to implement, we have also attempted this in previous experiments. Firstly, the detection results of the two methods were consistent. However, in practical operations, it is not feasible to collect a large amount of serum from kittens of younger ages, which means that there is insufficient serum for neutralization experiments. Therefore, we adopted the method described in the manuscript as the primary means for screening cats.

208 – challenge

Response: Thank you for pointing this out. We deleted the “challenge”. Revisions have been made in the manuscript (line 223).

202 – says 39. 3 x 14 = 42.

Response: Thank you for pointing this out. In this study, a total of 39 cats were used for the experiments, divided into 13 groups. The error has been corrected in the manuscript.

210-216 is repetitive with the preceding section.

Response: Thank you for pointing this out. We have reorganized the language at the corresponding locations in the manuscript, removed redundant statements, and introduced the immunization dosage and method. Your suggestions have made the article more rigorous. Lines 223-224.

218 says “second” – but that is not described above.

Response: immunization strategy in the manuscript, emphasizing the initial immunization and the booster immunization. Lines 227-231.

230 – I guess you should say how you quantified the virus

   Response: Thank you for pointing this out. We have supplemented the detection method. Using the FCV-fluorescence quantitation method developed by our laboratory to detect virus loads in viral excretion and viremia. This method belongs to a one-step quantitation method, which has extremely high accuracy and sensitivity. However, since the article describing this method has not been published yet, it has not been elaborated in detail in the manuscript of this study.

2.8 challenge test – were any of the observers blinded to the groups the cats were in?

Response: To ensure the rationality of the experiment and minimize the result bias caused by subjective factors, the challenge test was conducted by staff from the experimental animal center, while we were responsible for observing the clinical symptoms. Observations were carried out in pairs, and all groups were uniformly represented by numbers until the final day when we made unified annotations. In the manuscript, we have also provided detailed supplements regarding the observation methods and sample handling procedures.

238 – add that these are obtained from China.

Response: We have already added inmanuscript. Line 264.

239 – phylogenetic analysis does not evidence prevalence. It does talk of diversity.

Response: We have removed the mention of "prevalence" as it was inaccurate. Our genetic evolutionary tree also includes information on species, time and geography, allowing us to observe the isolation locations of FCV strains in China. Lines 276.

Line 240 is repetitive with what follows

Response: Thank you for pointing this out. We have removed redundant information in the manuscript. Lines 264-269.

244 – there are no leopard sequences in the tree.

Response: In Figure 2, there are sequences from three cheetahs. We summarized the information based on the specific host data provided on NCBI, which may result in some differences in the naming compared to leopards. Lines 276.

254 – the red and blue dot do not help…. and are not needed – they are easy to see.

Response: Thank you for pointing this out. We have removed the content from Figure 2 accordingly. Lines 276.

260 – personally I think these data including table 1 are better suited to the methods

Response: Based on the suggestions from various instructors, we have moved Tables 1 and 2 to the supplementary materials, which has made the article flow more smoothly.

Figure 2 – are the two strains used in your experiments also in this phylogeny. Assuming they are they should be highlighted clearly. To simplify the figure consider removing “china” from each name.

 Response: Thank you for pointing this out. We have annotated the virus strains DL39 (MW804430) and FB-NJ-13 (KM111557) used in this experiment with red and blue spots in Figure 2, which allows readers to see them more clearly. Additionally, we have provided explanations in the figure legend. Lines 276.

Line 261-262 are not needed. Indeed you could probably start this section around line 262.

Response: Thank you for pointing this out. We have deleted the sentence that did not contribute to the content. Line 286.

Line 264-5 can go – it is methods

Response: Thank you for pointing this out. We have made supplements to the Methods section in the manuscript. Lines 131-133.

Figure 3 – are all three panels needed?

Response: We believe that all three images are necessary. Firstly, we conducted SDS-PAGE verification to estimate the correctness of the bands. After that, we performed specificity verification for each protein to ensure that the expressed proteins are required for subsequent experiments. Therefore, the verification results in Figure 3 are essential for our follow-up experiments.

Table 2 could also be removed – you could just say concentrations ranged from x-y.

Response: Based on the suggestions from various instructors, we ultimately decided not to delete Table 2, but we have moved it to the supplementary materials, which has made the article flow more smoothly.

295 – reactogenicity is not a word I know. Sorry.

Response: Thank you for pointing this out. We have replaced it with "reactivity" in the manuscript. Line 316.

Section 3.3 – this can be simplified by dealing with both strains together. In both caess, CE had the most profoud effect. F and BD were also significant.

Response: Thank you for pointing this out. In our manuscript, we have consolidated the presentation of the two sections into one. Lines 309-314.

311 and elsewhere (eg line 321) – I am not sure it is right to describe these antigens as having “neutralising activity”.

Response: Thank you for pointing this out. In our manuscript, we have revised the terminology to "neutralizing antigen," a well-recognized definition in the field. Lines 341-342.

358 – perhaps remind us of the strain that Miaosanduo is based on?

Response: The strain utilized in Miaosanduo is the FCV-255 strain, which has been previously introduced and thus is not described again here.

359 – if comparing probably should add a statistical test for significance?

   Response: Thank you for pointing this out. We have incorporated statistical test for significance into the figure. Line 382.

363 – I am not sure you can call them “epidemic” strains.

Response: Thank you for pointing this out. The term "epidemic" was deemed inaccurate in our context, and we have substituted it with "diverse" in the manuscript to convey our meaning more precisely. Line 384.

369 – all challenged groups – I think you mean those not vaccinated?

Response: We have changed it to the "positive group" for easier understanding. Lines 390, 417, 443.

378 – figure 9b, c does not talk about ulcers

Response: Thank you for pointing this out. In our manuscript, we have moved Figure 9b, c to their respective appropriate positions and have reviewed and revised the subsequent two paragraphs for any similar descriptions to ensure consistency and accuracy. Lines 398, 425, 451.

379 – they are nearly all challenge groups. Use the group names you defined earlier, perhaps put in brackets (control)

Response: Thank you for pointing this out. We have made modifications in the manuscript by renaming the "challenged group" as the "positive group," and have also provided detailed descriptions of the various immune groups to enhance clarity for readers in understanding our work.

230 – describe method of quantification of virus

Response: In our manuscript, we have supplemented this information, and a detailed explanation has been provided in response to your previous question numbered 230.

378 – you have not measured virus shedding as this is a genome-based assay. In my opinion, a simple virus isolation may have been more informative but not necessary.

Response: Thank you for pointing this out. Your suggestion is extremely valuable. In our actual operations, we have indeed conducted similar procedures, albeit on a subset of samples. The results confirmed the presence of viable viruses, however, this method lacks precision in quantifying the virus. We attempted to directly measure virus titer using the collected samples, but the results were not satisfactory. Therefore, in this particular experiment, we primarily relied on fluorescence quantification to determine virus content. In our subsequent animal experiments, we will consider combining this method with fluorescence quantification. Thank you for your invaluable advice.

Figure 9a – could this be coloured the same as the rest of the figure? And please explain in the methods probably what you have done to the recorded clinical scores to produce this overall score.

Response: Thank you for pointing this out. We also attempted to modify the colors to correspond with different groups for easier observation. However, the selection of image colors indeed necessitates a balance among multiple factors, including the clarity of information, the visual effect of the image, and the reading experience of the audience. After the modification, the image became more prominent due to the oversized column volumes, and the specific information was not fully highlighted. We adopted a black color scheme with red fonts for scores, as this may render the information presented in the image clearer. Therefore, we would like to consult with you on whether we can retain the original color scheme. Detailed observation records have been presented in the methodology, and clinical scoring items and scores have been displayed in Table S3. Lines 246-251, Table S3.

448 – you have two control groups. Non-challenge control group. Again use the group designations you defined earlier.

Response: Thank you for pointing this out. We have made modifications in the manuscript to clearly distinguish between the various groups.

411 – delete “extremely high”

Response: Thank you for pointing this out. In the manuscript, we have made deletions accordingly. Lines 432.

414 – 415 – not sure you can defend “100% protection”. Remove this last sentence

Response: Thank you for pointing this out. In the manuscript, we have made revised accordingly. Lines 409, 435, 460.

What has happened to the results of the HRB48 challenge.

Response: The results of HRB48 (GI) challenge is the recombinant proteins CE39 and CE39-CEFB confer a effective protection rate against the genotype GI strain HRB48 than Miaosanduo.

452 – challenged groups is again misleading. I think you mean unvaccinated controls.

Response: Thank you for pointing this out. In the manuscript, we have made revised accordingly.

Figure 12 legend needs improving – the bottom six panels I think are unvaccinated controls. It might be better if the column was the challenge stain, and the row whether or not the cat was vaccinated. The arrows do not help. The entire view shows abnormal pathology (although I am not a pathologist). Use the arrows to point to more specific things described in the text.

Response: Thank you for pointing this out. We have made modifications based on your suggestions. Firstly, we have switched the horizontal and vertical arrangements. Secondly, after consulting with a pathology instructor, we confirmed that the pathological sections are correct. However, we identified some errors in the arrow annotations for the positive group sections. We have re-annotated them according to the discussion outcomes. Line 481.

475 – domestic Chinese

Response: Thank you for pointing this out. Revisions have been made in the manuscript (line 496).

477-479 – I sense trivalent is misleading here. I don’t think there is evidence that vaccinated cats are at increased risk is there? And vaccines are not licensed to protect chronic stomatitis. I guess it depends what you mean by “stomatitis”. Think this sentence needs careful thought.

Response: Thank you for pointing this out. Revisions have been made in the manuscript (lines 496-498).

527 – 535 – can be removed. Or just briefly mentioned as future work. Your experiments did not (sorry if I missed it) include a non-adjuvanted control so you don’t even know if the addition of an adjuvant had an effect. Perhaps this should Be mentioned as a limitation.

Response: Thank you for pointing this out. We have deleted that particular section and rewritten the discussion accordingly.

Line 543 onwards is conclusion. This can be considerably shortened.

Response: Thank you for pointing this out. In the manuscript, we have revised the conclusion accordingly. Line 616.

Discussion – for me the discussion lacks focus and is disappointing. Interesting things you could have considered – is it usual for vaccinated cats to shed post-challenge, did the 39/fb recombinant improve protection, what about a particular strain makes it broadly cross-reactive, could the conserved regions in the peptides be contributing to cross-reactivity, do you feel your peptides contain relevant conformation, or are they more likely to be linear. Perhaps their results could be compared to other caliciviruses like human norovirus or RHDV where I suspect this approach has been more considered. Indeed there is a recombinant myxo:RHDV vaccine on the market and as for the human noroviruses,  culturing is not an option. Adding this human dimension would also broaden the interest of this paper. Indeed it might be worth stating in the introduction that this antigen approach has been tried for a range of other viruses especially human norovirus.  I am sure the authors can think of other discussion points. The large section on adjuvants seems largely irrelevant.

Response: Thank you for pointing this out. We have rewritten the discussion accordingly. Lines 523-537, 549-615.

Reviewer 3 Report

Comments and Suggestions for Authors

The authors described the design of a subunit vaccine against feline calivirus based on the CDE subunit of the capsid protein VP1 and the combination of subunits derived from two different strains to increase cross-protection. The study is based on in vitro assays and infectious challenge experiments.

General comments:

-          The article should be reviewed by a native English speaker

-          The fact that the CDE region contains the main epitopes involved in the protective immune response was already known. Epitope mapping has been done by several groups and a recent article (Li et al. BMC Veterinary Research - https://doi.org/10.1186/s12917-024-03914-2) has reported the use of the CDE protein subunit for a FCV vaccine. The originality of this manuscript lies more in the combination of the two CDE subunits from DL39 and FB-NJ-13 strains. The discussion should then be revisited and focused on what the manuscript brings as novelty: the use of CDE subunit as vaccine candidate and potential combinations between subunits from different strains.

Detailed comments :

Abstract :

-          Line 20 : consistently with the general comments, the authors should state that they confirmed that CDE region was the optimal protective region, instead of “pintpointed”. This has already been demonstrated. More generally, several studies have been published on the mapping of epitopes involved in neutralization and have shown that most of them are located in region E and to a lesser extent in region C.

Introduction :

-          Line 64 : the authors suggest there is an “epidemic trend in the country”. In most countries, FCV is usually endemic in the cat population. It is rather an endemic infection with an increased awareness due to more frequent investigations and surveys, especially in China.

-          Line 66: the authors wrote that “this disease has rapidly spread”. In fact, there has been foci of highly virulent FCV infection which occurred independently. The word “spread” is misleading here.

-          Lines 76-78: lack of inactivation with killed FCV vaccines is not a reported issue. The authors should clarify what they mean by “caused by the incomplete inactivation of some attenuated vaccine viruses”. Attenuated vaccines are not inactivated. It seems that there is confusion between incomplete inactivation (killed vaccines) and reversion to virulence (attenuated vaccines). Some residual virulence has been reported with FCV F9 attenuated FCV strain and several articles reported FCV cases due to FCV-F9-like strains.

Methods:

-          Lines 108-119: figure S1(b) is a nice illustration of the position of the different regions within VP1 overall structure. However, the authors should clearly write that nothing has been done to check the structure of the different subunits after expression. It is likely that the structure of the subunits may be somewhat different from their original structure within VP1.

-          Lines 120-131: the authors should precise which bacterial strain they used.

-          Lines 133-151: no washing step is described in the competitive ELISA method. It would be useful to describe them.

-          Lines 216-217: instead of DMEM, the authors could have used the Montanide gel. Could the authors comment on why they didn’t use the adjuvant as a placebo.

Results:

-          Table 1 & Table 2: these data might be put in supplementary materials since they don’t directly participate to the conclusions of the article.

-          Lines 255-270: beyond the purity of expressed proteins, it would have been interesting to test the ability of a panel of characterized monoclonal antibodies to recognize some key epitopes. One of the limits of this approach is that we don’t know the impact of the truncation into subunits on the antigenicity of those subunits compared to that of VP1.

-          Fig.4: region E contains immunodominant hypervariable domains with neutralizing epitopes. Surprisingly, region E had a low competitive activity in the ELISA (similar to region B). Could the authors explain why?

-          Fig.5: the E subunit had a stronger competitive activity in the neutralization assay and this result is more conform to expectations. Could the authors explain this discrepancy between ELISA and neutralization competitive assays for subunit E?

-          Fig.6: it is interesting to see that for HRB48 (GI), the neutralization competitive activity is higher for CEfb (GII) than for DL39 (GI). It raises the question of whether genotype is correlated with cross-neutralization. In other words, will a strain from one genotype better cross-neutralize the strains from the same genotype than the strains of the other genotype? This should be addressed in the discussion.

-          Fig.7: CE39-CEfb has an apparent MW of 50-55kD? While CE39 and CEfb have MW around 33-35kD (Fig.3). Shouldn’t we expect a MW around 66kD for the combination? How do you explain this?

-          Fig.8: why didn’t the authors include a CEfb immunization group? According to fig.6, CEfb seems to have a stronger cross-neutralizing inhibition activity.

-          Lines 387-389: could the authors explain the “100% of protection claim” while clinical signs, lower weight gain than negative controls, viremia and viral excretion were reported in all challenged groups.

-          Lines 414-415: same comment

Discussion:

-          Lines 524-530: addition of cytokines to enhance the efficacy of vaccines is indeed a theoritical attractive concept but it raises significant safety and regulatory questions when it comes to develop and register the vaccine for commercial use. In addition, the use of adjuvants in feline vaccines is a controversial issue. In some cats, severe adverse reactions, including injection-site sarcomas, have been reported after vaccination.

-          Lines 530-536: the list of adjuvants would be more appropriate in the writing of a patent than in this article. It has no added value in this discussion.

-          Lines 536-542: those lines address a more relevant topic which should be discussed further. The authors should comment on the fact that CE subunits might conserve their linear neutralizing epitopes. However what about conformational epitopes?

-          The authors should use the discussion to give some explanations on the discrepancies between ELISA and neutralization assay results.

-          The authors should also discuss whether the combination of CE39 and CEfb is the best one to provide broad cross-protection in the field. The challenge experiments did not show a clear difference.

-          It would be useful to write about the limitations of the study, e.g. expression of subunits may impact the antigenicity of the different regions, notably the conformational epitopes.

-          Apart from challenge with FB-NJ-13 (where it is logical that CE39-CEfb performs better than CE39), the difference of protection between CE39 and CE39-CEfb is not obvious. Is CE39 and CEfb the best combination in terms of complementarity?

Conclusion:

lines 560-562: none of the vaccines provided 100% of protection. Vaccines were able to reduce clinical sign and viral excretion, but did not provide full protection. This statement should

Comments on the Quality of English Language

The manuscript should be reviewed by an English speaker

Author Response

I deeply appreciate the time and effort you have dedicated from your hectic schedule to offer valuable guidance on my paper. We have diligently responded to each of your suggestions and incorporated the necessary corrections into the manuscript. Your insightful advice has played a pivotal role in improving the fluency, coherence, and rigorousness of our paper. I extend my heartfelt best wishes to you. The article has been reviewed by a native English speaker and primarily rewritten the discussion. My specific answers as follows:

Comments and Suggestions for Authors

The authors described the design of a subunit vaccine against feline calivirus based on the CDE subunit of the capsid protein VP1 and the combination of subunits derived from two different strains to increase cross-protection. The study is based on in vitro assays and infectious challenge experiments.

General comments:

- The article should be reviewed by a native English speaker

Response: Thank you for pointing this out. We agree with this comment and the article has already been reviewed by a native English speaker.

- The fact that the CDE region contains the main epitopes involved in the protective immune response was already known. Epitope mapping has been done by several groups and a recent article (Li et al. BMC Veterinary Research - https://doi.org/10.1186/s12917-024-03914-2) has reported the use of the CDE protein subunit for a FCV vaccine. The originality of this manuscript lies more in the combination of the two CDE subunits from DL39 and FB-NJ-13 strains. The discussion should then be revisited and focused on what the manuscript brings as novelty: the use of CDE subunit as vaccine candidate and potential combinations between subunits from different strains.

Response: Although Li et al. have affirmed that the CDE region is the principal antigen region of FCV, it is still indispensable to screen a suitable strain that can offer broad-spectrum immunity. Our previous research has verified that DL39 could provide broad-spectrum protection against the infection of numerous FCV strains, and now we have confirmed that the combination of DL39 and the CDE region of FB-NJ-13 could enhance the immunological protection of vaccines. Revisions have been carried out in the manuscript. Lines 560-570, 590-603.

Detailed comments:

Abstract:

- Line 20 : consistently with the general comments, the authors should state that they confirmed that CDE region was the optimal protective region, instead of “pintpointed”. This has already been demonstrated. More generally, several studies have been published on the mapping of epitopes involved in neutralization and have shown that most of them are located in region E and to a lesser extent in region C.

Response: Thank you for pointing this out. We agree with this comment. Revisions have been made in the manuscript. Lines 19.

Introduction:

- Line 64 : the authors suggest there is an “epidemic trend in the country”. In most countries, FCV is usually endemic in the cat population. It is rather an endemic infection with an increased awareness due to more frequent investigations and surveys, especially in China.

Response: Thank you for pointing this out. We agree with this comment. Revisions have been made in the manuscript. Lines 60.

- Line 66: the authors wrote that “this disease has rapidly spread”. In fact, there has been foci of highly virulent FCV infection which occurred independently. The word “spread” is misleading here.

Response: Thank you for pointing this out. We agree with this comment. Revisions have been made in the manuscript. Lines 63.

- Lines 76-78: lack of inactivation with killed FCV vaccines is not a reported issue. The authors should clarify what they mean by “caused by the incomplete inactivation of some attenuated vaccine viruses”. Attenuated vaccines are not inactivated. It seems that there is confusion between incomplete inactivation (killed vaccines) and reversion to virulence (attenuated vaccines). Some residual virulence has been reported with FCV F9 attenuated FCV strain and several articles reported FCV cases due to FCV-F9-like strains.

Response: Thank you for pointing this out. We agree with this comment. The relevant descriptions have been eliminated on account of inaccuracy.

Methods:

- Lines 108-119: figure S1(b) is a nice illustration of the position of the different regions within VP1 overall structure. However, the authors should clearly write that nothing has been done to check the structure of the different subunits after expression. It is likely that the structure of the subunits may be somewhat different from their original structure within VP1.

Response: Thank you for pointing this out. We agree with this comment. Revisions have been made in the manuscript. Lines 113-115.

- Lines 120-131: the authors should precise which bacterial strain they used.

Response: Thank you for pointing this out. BL21 competent cells were used to express the recombinant Proteins. Lines 113-115.

- Lines 133-151: no washing step is described in the competitive ELISA method. It would be useful to describe them.

Response: Thank you for pointing this out. Revisions have been made in the manuscript. Lines 139.

- Lines 216-217: instead of DMEM, the authors could have used the Montanide gel. Could the authors comment on why they didn’t use the adjuvant as a placebo. 

Response: Thank you for pointing this out. The positive control group and negative control group received the same volume of the mixture DMEM with Montanide™ GE (1:1) for immunization. Lines 227-229.

Results:

- Table 1 & Table 2: these data might be put in supplementary materials since they don’t directly participate to the conclusions of the article.

Response: Thank you for pointing this out. We agree with this comment and the two tables have been put in supplementary materials.

- Lines 255-270: beyond the purity of expressed proteins, it would have been interesting to test the ability of a panel of characterized monoclonal antibodies to recognize some key epitopes. One of the limits of this approach is that we don’t know the impact of the truncation into subunits on the antigenicity of those subunits compared to that of VP1.

Response: Thank you for pointing this out. Due to the lack of available monoclonal antibodies which can recognize all different regions of VP1, it’s hard to assess the antigenicity between those subunits and VP1. But we believe those subunits could stimulate the cats to produce high levels of antibodies based on the neutralization ability of sera from immunized cats.

- Fig.4: region E contains immunodominant hypervariable domains with neutralizing epitopes. Surprisingly, region E had a low competitive activity in the ELISA (similar to region B). Could the authors explain why?

Response: Thank you for pointing this out. Because competitive ELISA can only detect the total antibodies level but cannot ascertain whether these antibodies exhibit neutralizing activity. Even though the antibodies level induced by region E is similar with those induced by region B, the antibodies induced by region E showed better neutralizing activity than those induced by region B according to the results of competitive neutralization tests. The antibodies induced by region CE showed better competitive activity than those induced by region B in the ELISA, which is the same with the results of competitive neutralization tests. Thus, we speculate that region E is less stable compared with region CE, which lead to the antigenicity insufficient of region E.

- Fig.5: the E subunit had a stronger competitive activity in the neutralization assay and this result is more conform to expectations. Could the authors explain this discrepancy between ELISA and neutralization competitive assays for subunit E?

Response: Thank you for pointing this out. It’s easy to detect the level of antibodies induced by subunit E with ELISA but impossible to judge if those antibodies have the ability of neutralization. 

- Fig.6: it is interesting to see that for HRB48 (GI), the neutralization competitive activity is higher for CEFB (GII) than for DL39 (GI). It raises the question of whether genotype is correlated with cross-neutralization. In other words, will a strain from one genotype better cross-neutralize the strains from the same genotype than the strains of the other genotype? This should be addressed in the discussion.

Response: Thank you for pointing this out. Usually, the strains form them same genotype have the better ability to produce antibodies to neutralize those in the same group rather than those form another genotype in Fig. 2S. Nevertheless, several VSD FCV strains don’t obey this regularity according to our previous studies. It still needs more studies to confirm this regularity. And HRB48 show higher neutralization competitive activity for CEFB because the sera produced by FB-NJ-13 shows much lower neutralization activity for HRB48 than the sera produced by DL39. Thus, the results of Fig.5 and Fig.6 can only confirm region CE is the potential dominant antigen epitopes but cannot judge the neutralization competitive activity between CE39 and CEFB. We also discussed this interesting phenomenon during our discussion. Lines 549-560.

- Fig.7: CE39-CEfb has an apparent MW of 50-55kD? While CE39 and CEfb have MW around 33-35kD (Fig.3). Shouldn’t we expect a MW around 66kD for the combination? How do you explain this?

Response: Thank you for pointing this out. Each protein was expressed with an antigen region, a TrxA tag and a His tag. CE39 or CEFB have MW around 20-21 kDa and a TrxA tag has MW around 13-14 kDa. CE39-CEFB has an apparent MW of 50-55 kDa thus. In the manuscript, we have also supplemented information regarding the vector and the components of various proteins to facilitate a better understanding for the readers. Line 128.

- Fig.8: why didn’t the authors include a CEfb immunization group? According to fig.6, CEfb seems to have a stronger cross-neutralizing inhibition activity.

Response: Thank you for pointing this out. According to our previous studies and the results of competitive ELISA assays and neutralization tests,DL39 strain shows better broad-spectrum immunity than others including FB-NJ-13 strain. CEFB seems to have a stronger cross-neutralizing inhibition activity because the sera produced by FB-NJ-13 have a lower neutralizing activity than the sera produced by DL39. Thus, we didn’t set the CEFB immunization group.

- Lines 387-389: could the authors explain the “100% of protection claim” while clinical signs, lower weight gain than negative controls, viremia and viral excretion were reported in all challenged groups.

Response: Thank you for pointing this out. We agree with this comment. In our conclusions, we have removed the phrase "100% protection rate," as this approach is indeed controversial. As you have pointed out, although the subunit vaccine demonstrates better efficacy than the currently marketed vaccine strain FCV-255, some clinical symptoms still manifest. Furthermore, our validation is limited to preliminary animal trials, rendering this conclusion inaccurate. However, our immunity group exhibited milder clinical symptoms, reduced virus excretion, and an absence of oral ulcerations. In the subsequent stages, we will continue to refine the evaluation of the vaccine. Your suggestion is invaluable for enhancing the rigor and accuracy of our manuscript. Thank you for your precious feedback. Specific modification location. line 409, 435, 460, 631, 635.

- Lines 414-415: same comment

Response: Thank you for pointing this out. We agree with this comment. Revisions have been made in the manuscript. line 409, 435, 460, 631, 635.

Discussion:

- Lines 524-530: addition of cytokines to enhance the efficacy of vaccines is indeed a theoritical attractive concept but it raises significant safety and regulatory questions when it comes to develop and register the vaccine for commercial use. In addition, the use of adjuvants in feline vaccines is a controversial issue. In some cats, severe adverse reactions, including injection-site sarcomas, have been reported after vaccination.

Response: Thank you for pointing this out. Although adjuvants or immune-enhancing factors can cause adverse reactions in extremely rare cats, the addition of these substances is necessary to improve the immune response of subunit vaccines. Therefore, our subsequent research should focus on developing immune adjuvants and additive factors that can induce higher levels of intermediate and antibody response, while being safe.

- Lines 530-536: the list of adjuvants would be more appropriate in the writing of a patent than in this article. It has no added value in this discussion.

Response: Thank you for pointing this out. Revisions have been made in the manuscript. The relevant descriptions have been eliminated on account of valuelessness.

- Lines 536-542: those lines address a more relevant topic which should be discussed further. The authors should comment on the fact that CE subunits might conserve their linear neutralizing epitopes. However, what about conformational epitopes?

Response: Thank you for pointing this out. The suboptimal immune effectiveness could potentially be ascribed to the fact that the amino acid peptide segments expressed by the prokaryotic system might possess disparate advanced structures in comparison with the natural virus particle, leading to the loss of partial or complete conformational antigenic epitopes, or the swift release of antigenic short peptides, which failed to induce higher antibody levels prior to degradation. Thus, the further optimization of subunit vaccines is requisite. Revisions have been made in the manuscript. Line 592-603.

- The authors should use the discussion to give some explanations on the discrepancies between ELISA and neutralization assay results.

Response: Thank you for pointing this out. Competitive ELISA demonstrates remarkable reproducibility and is ideally methods for the identification and screening of viral antigen epitopes, having already been implemented in practical applications. However, that ELISA can only quantify the total antibody levels produced and cannot ascertain whether these anti-bodies exhibit neutralizing activity. For instance, this study employed competitive ELISA to illustrate that the BD, CE, and F regions of the FCV structural protein VP1 in both the DL39 and FB-NJ-13 strains exhibit significant immunogenicity, leading to the induction of high antibody titers. Nevertheless, it remains uncertain whether these antibodies possess neutralizing activity. Neutralization assays can be utilized to detect the neutralizing antibody levels produced by the body, and competitive neutralization as-says can also determine the specific antigenic regions or peptide segments that generate neutralizing antibodies. Revisions have been made in the manuscript. Line 523~537.

- The authors should also discuss whether the combination of CE39 and CEfb is the best one to provide broad cross-protection in the field. The challenge experiments did not show a clear difference.

Response: Thank you for highlighting this. Our previous findings have incontrovertibly demonstrated that the FCV DL39 strain exhibits robust broad-spectrum protective efficacy against the currently prevailing genotype GI and GII strains. Moreover, the combination of CE39 and CEFB will enhance the broad-spectrum property of vaccines. Nevertheless, the outcomes of the challenge experiments might have been unobvious as a result of effective immunogenicity of single region CE39. Line 541-548.

- It would be useful to write about the limitations of the study, e.g. expression of subunits may impact the antigenicity of the different regions, notably the conformational epitopes.

Response: Thank you for pointing this out. Revisions have been made in the manuscript. Line 592-603.

- Apart from challenge with FB-NJ-13 (where it is logical that CE39-CEfb performs better than CE39), the difference of protection between CE39 and CE39-CEfb is not obvious. Is CE39 and CEfb the best combination in terms of complementarity?

Response: Thank you for pointing this out. At the current stage of research, we cannot yet conclude that our combination is the optimal protective antigen combination, as assessing whether a combination is optimal typically requires comparison and validation against multiple other combinations. Our combination principle is based on antigens with broad-spectrum characteristics, which thus possesses a certain degree of rationality. Based on the current experimental results, we have successfully identified a protective antigen combination pattern that performs well across different genotypes. To further enhance our research findings, we plan to conduct further validation by combining the best protective antigens from different virus strains in the future. Revisions have been made in the manuscript. Lines 541-548, 565-570.

Conclusion:

lines 560-562: none of the vaccines provided 100% of protection. Vaccines were able to reduce clinical sign and viral excretion, but did not provide full protection. 

Response: Thank you for pointing this out. Thank you for pointing this out. We agree with this comment. In our conclusions, we have removed the phrase "100% protection rate," as this approach is indeed controversial. Revisions have been made in the manuscript. Lines 633-636.

Round 2

Reviewer 1 Report

Comments and Suggestions for Authors

I think that the authors have adequately addressed the comments made by me in the revised version of the manuscript. Therefore, I have no further comments and consider the paper should accepted for publication.

Author Response

Once again, I convey my profound appreciation for your expert guidance and invaluable assistance. With sincerest regards and warmest wishes extended to you.

Reviewer 3 Report

Comments and Suggestions for Authors

The authors have addressed the comments and adequately revised their manuscript. A few typos remain:

-          Line 117: typo “structure”

-          Line 141: typo “sera”

-          Line 312: the authors should avoid words like “extremely” in a scientific article. Writing that the difference is statistically significant and giving the p value is clear enough.

-          Lines 331 & 335: “the better neutralizing antigen” should be replace by “the main targets of neutralizing antibodies”

-          Line 460: typo “demonstrated”

Author Response

I am deeply grateful for the valuable time and effort you dedicated from your hectic schedule to re-examine my paper and highlight the errors contained within. We have diligently addressed and corrected the mistakes identified in the manuscript. The following section presents the specific responses to the key issues you raised, along with the revisions we have implemented:

  1. Line 117: typo “structure”

Response: Thank you for pointing this out. We agree with this comment. Revisions have been made in the manuscript. Lines 117.

  1. Line 141: typo “sera”

Response: Thank you for pointing this out. We agree with this comment. Revisions have been made in the manuscript. Lines 141.

  1. Line 312: the authors should avoid words like “extremely” in a scientific article. Writing that the difference is statistically significant and giving the p value is clear enough.

Response: Thank you for pointing this out. We agree with this comment. Revisions have been made in the manuscript. Lines 256-259, 311.

  1. Lines 331 & 335: “the better neutralizing antigen” should be replace by “the main targets of neutralizing antibodies”

Response: Thank you for pointing this out. We agree with this comment. Revisions have been made in the manuscript. Lines 330 & 334.

  1. Line 460: typo “demonstrated”

Response: Thank you for pointing this out. We agree with this comment. Revisions have been made in the manuscript. Lines 459.